



# Star photometry with all-sky cameras to retrieve aerosol optical depth at night-time

Roberto Román[1,2], Daniel González-Fernández[1,2], Juan Carlos Antuña-Sánchez[1,3], Celia Herrero del Barrio[1,2], Sara Herrero-Anta[1,2], África Barreto[4,1], Victoria E. Cachorro[1,2], Lionel Doppler[5], Ramiro González[1,2], Christoph Ritter[6,7], David Mateos[1,2], Natalia Kouremeti[8], Gustavo Copes[9], Abel Calle[1,2], María José Granados-Muñoz[10,11], Carlos Toledano[1,2], and Ángel M. de Frutos[1,2]

[1]Group of Atmospheric Optics (GOA-UVa), Universidad de Valladolid, 47011, Valladolid, Spain
[2]Laboratory of Disruptive Interdisciplinary Science (LaDIS), Valladolid, Spain
[3]GRASP-SAS, Lille, France
[4]Izaña Atmospheric Research Center, Meteorological State Agency of Spain (AEMet), Spain
[5]Deutscher Wetterdienst, Meteorologisches Observatorium Lindenberg – Richard-Assmann-Observatorium (DWD, MOL-RAO), Lindenberg (Tauche), Germany
[6]Institute of Physics and Astronomy, University of Potsdam, 14476, Potsdam, Germany
[7]Alfred-Wegener-Institute Helmholtz Centre for Polar and Marine Research, 14473, Potsdam, Germany
[8]Physikalisch-Meteorologisches Observatorium Davos and World Radiation Center (PMOD/WRC), Davos, Switzerland
[9]Servicio Meteorológico Nacional, Argentina
[10]Department of Applied Physics, Universidad de Granada, 18071, Granada, Spain
[11]Andalusian Institute for Earth System Research, IISTA-CEAMA, Granada, Spain

**Correspondence:** Roberto Román (robertor@goa.uva.es)

**Abstract.** The lack of aerosol optical depth (AOD) data at night can be partially addressed through moon photometer measurements or fully covered with star photometer observations. However, the limited availability and complexity of star photometers has motivated this study to use all-sky cameras to extract starlight signals and derive AOD at night using star photometry. For this purpose, eight all-sky cameras were configured and deployed in nine different locations to capture raw images with varying

exposure times every 2 minutes during the night. This work proposes a novel methodology to extract the starlight signal from the raw data of all-sky cameras and convert it into AOD values. This process consists of the following steps: removing the background image, selecting the pixels and extracting the signal for each star from a predefined list of 56 stars, performing in-situ Langley calibration of the instruments and retrieving the total optical depth (TOD), calculating the effective wavelength for each camera channel, deriving the AOD by subtracting the gas contribution to TOD, and averaging, cloud-screening, and

quality-assuring the AOD time series. The AOD time series obtained through this methodology are compared with independent AOD measurements from collocated moon photometers in the nine locations. The obtained results show that the AOD values derived with the proposed method generally correlate with reference values, often achieving correlation coefficients (r) above 0.90. The AOD values retrieved using the cameras tend to overestimate the reference values by approximately 0.02, and exhibit a precision of around 0.03-0.04. The agreement between both datasets varies with wavelength and decreases at high-latitude

locations, likely due to the poorer performance of Langley calibration in these regions. AOD values align well with day-to-night transitions obtained by solar photometers, demonstrating their reliability. Despite the slight overestimation, the AOD values





derived by this new method approximate the real values and provide coverage throughout the entire night, without requiring the presence of the Moon. Therefore, they serve for studying and monitoring the nocturnal evolution of AOD.

## 1 Introduction

The study of aerosol optical properties and their temporal variability is essential to better understand their role in the Earth's climate system. Aerosols influence radiative forcing both directly, by scattering and absorbing solar radiation, and indirectly, by modifying cloud properties and lifetimes (Boucher et al., 2013). This is particularly critical in polar regions, where aerosols can have a significant influence due to the unique atmospheric and surface conditions (Law and Stohl, 2007). In these regions, the high albedo of ice and snow amplifies the impact of aerosols on the radiative balance, and the deposition of light-absorbing

particles, such as black carbon, can accelerate ice and snow melt (Hansen and Nazarenko, 2004; Bond et al., 2013). It is therefore essential to continuously monitor aerosol properties and generate long-term data series to study potential changes and trends, especially in polar regions, which remain underrepresented in global aerosol monitoring due to logistical challenges and harsh environmental conditions.

One of the most important aerosol properties is the aerosol optical depth (AOD), which is related to the extinction (scattering

+ absorption) of light caused by aerosols as it crosses the atmosphere. AOD serves as a proxy for the amount of aerosol particles present in the atmosphere. AOD spectral variation, quantified through the empirical Ångström's law (Angström, 1961), also provides information about particle size predominance. AOD is typically calculated using measurements from photometers that, by directly pointing at the Sun, measure the direct solar irradiance reaching the Earth's surface at one or generally multiple wavelengths separately. If the photometer is properly calibrated, these direct irradiance measurements can

be converted to AOD values through Beer–Bouguer–Lambert law. One method to calibrate photometers for obtaining AOD values is the technique named "Langley" or "Langley-plot" (Shaw, 1983). This method involves performing direct irradiance measurements at different solar zenith angle (SZA) values, fast enough to assume that the direct extraterrestrial solar irradiance and the atmospheric optical depth remain constant (Toledano et al., 2018). It allows to calculate AOD without knowing the true extraterrestrial solar irradiance in physical units. This methodology is followed by AERONET (AErosol RObotic NETwork;

Holben et al., 1998, https://aeronet.gsfc.nasa.gov), a network which includes more than 500 photometer stations distributed worldwide. AERONET calibrates its field photometers by intercomparison with reference photometers (masters), which are themselves calibrated using the Langley technique (Giles et al., 2019).

Thanks to this method and direct solar irradiance measurements, AOD can be obtained during the daytime, but not at night-time. However, knowing the AOD at night is crucial, among other issues, to study aerosol complete cycles (Perrone et al.,

2022; Herrero del Barrio et al., 2023) and the aerosol interaction with longwave radiation, especially in polar regions, where continuous night periods occur for several consecutive months during the polar night. Fortunately, some photometer models, such as those used in AERONET, were updated a few years ago to measure direct lunar irradiance at ground from the first moon quarter to the third one (Berkoff et al., 2011; Barreto et al., 2013). The main challenge with these measurements is that the direct extraterrestrial lunar irradiance changes significantly, even within a single night, making the Langley technique insufficient;



it is then necessary to know the direct extraterrestrial lunar irradiance each time. In this regard, the ROLO model (Robotic
Lunar Observatory; Kieffer and Stone, 2005), in the case of AERONET, or its implementation RIMO (ROLO Implementation
for Moon photometry Observation; Barreto et al., 2019), have been used to estimate the direct extraterrestrial lunar irradiance.
These models did not initially provide accurate AOD values (Barreto et al., 2016, 2017), but this issue was resolved after
applying corrections to them (Román et al., 2020; González et al., 2020; Uchiyama et al., 2019). However, this does not

address another issue with lunar photometry: there are no direct lunar irradiance measurements during half of the lunar cycle
(from the third quarter to the first one), and even during the other half, the Moon is not visible throughout the entire night,
especially near the quarter phases. This means that lunar photometry only partially fills the gap in our knowledge of night-time
AOD.

This gap in night-time data series can be filled by using star photometers. These instruments essentially consist of a telescope

on a tracking mount equipped with a sensor that records direct star irradiance at ground. AOD can be derived from these
measurements by applying star photometry (see Ivǎnescu et al., 2021, and references therein), which works similarly to solar
or lunar photometry but covers the entire night, as there are always multiple stars visible in the sky. Moreover, the direct
extraterrestrial irradiance of stars is highly stable over time, allowing the Langley technique to be applied without requiring
prior knowledge of these extraterrestrial values. In addition to Langley calibration, the capability of star photometers to measure

direct irradiance from multiple stars enables the application of alternative calibration methods, such as the two-star method,
which uses two different stars at varying zenith angles (Leiterer et al., 1995; Pérez-Ramírez et al., 2008), or the multi-star
Langley method proposed by Ivǎnescu and O'Neill (2023). Although star photometer measurements are still used to study
AOD at night (e.g., Graßl et al., 2024), the availability of star photometers is extremely limited due to, among other issues,
their high cost and the fact that they are not fully automated. There are currently only about five star photometers in operation

worldwide for aerosol monitoring (Barreto et al., 2019; Román et al., 2020).

This scarcity of star photometers has led to the search for alternatives that can be more widely used to obtain AOD at
night. The fact that the sensor of one of the existing star photometers is a CCD camera that captures images where the star
occupies only a small fraction of the CCD image (Pérez-Ramírez et al., 2008), suggests that other instruments capable of
capturing images with visible stars could also be used for star photometry. All-sky cameras can be used for this purpose. These

instruments typically consist of a CMOS sensor with a fisheye lens that captures images of the entire sky dome, as opposed to
focusing on a smaller magnified area like star photometers. Most modern all-sky cameras can operate both day and night, and
their wide field of view allows them to capture stars, which occupy a very small area in the image but include a large number
of stars as they observe the entire sky at once. Therefore, if the starlight can be extracted from these images, the night-time
AOD can be derived in a similar way to that of a star photometer. In fact, it has already been demonstrated that AOD can

be estimated during daytime using all-sky cameras with different methods (Cazorla et al., 2008; Román et al., 2022; Scarlatti
et al., 2023). Among its advantages, all-sky cameras are fully automated instruments, with no moving parts, and are much more
cost-effective.

In this framework, our starting hypothesis is that the starlight signal, proportional to star irradiance, can be extracted from
all-sky camera images, and thus, AOD can be obtained using star photometry. The main idea is that while the signal from each



individual star—and consequently the AOD derived from it—may be very noisy due to technical limitations of all-sky cameras, the ability to capture the signal from many stars simultaneously and to take continuous images allows for a large number of AOD values to be obtained. Although each individual value may have high noise, averaging them reduces the overall noise. With this in mind, the main objective of this work is to develop a new methodology capable of extracting the starlight signal from images captured by all-sky cameras and calculating AOD from these values. Additionally, we aim to test and validate

this new method by comparing the results with independently obtained measurements using lunar photometry across various locations.

     This paper is structured as follows: Section 2 introduces the technical characteristics and configuration of the all-sky cameras used in this study, as well as the locations where they are installed. The proposed methodology for extracting star direct irradiances and converting them into AOD values is detailed in Section 3. Section 4 presents a general comparison between

the AOD retrieved from the all-sky cameras and the values independently obtained by moon photometers, along with two case studies. Finally, the main conclusions are summarized in Section 5.

## 2   Instrumentation and Sites

The Group of Atmospheric Optics of the University of Valladolid (Grupo de Óptica Atmosférica; GOA-UVa) has created and developed GOA-SCAN (GOA all-Sky CAmeras Network), a network of all-sky cameras deployed around the world.

GOA-SCAN is managed by GOA-UVa, which establishes protocols for measurements, data storage, calibration, and quality assurance to maximize the information recorded by the cameras and to standardize and process the data consistently. This network comprises all-sky cameras from GOA-UVa as well as from other research groups.

     One of the most commonly used camera models in GOA-SCAN is the OMEA-3C all-sky camera, manufactured by *Alcor System*. This camera features a fisheye lens (180ºx180º, equidistant projection) mounted over a CMOS sensor (*SONY IMX178*)

with 3096 x 2080 pixels (6.44 megapixels allowing a resolution of 5.4 arcmin/pixel) and 14-bit resolution. This sensor has three distinct RGB filters (Red, Green, and Blue; see Figure 1a) arranged in an RGGB Bayer pattern mosaic, along with an additional IR-cut filter (see Figure 1b) placed over the RGB filters to block infrared skylight. The spectral response of the RGB filters, the IR-cut filter, and the combined effect of both (the total spectral response of each of the three RGB channels) are shown in Figure 1d. In GOA-SCAN, there is a modified version of the OMEA-3C camera that is identical to the original

but replaces the infrared filter with a tri-band filter (see Figure 1c), which narrows the spectral bandwidth of the three color channels (see Figure 1e). We will refer to this modified camera version as OMEA-3C-TF (TF for tri-band filter). In this work, we have also used grayscale (Gr) images, which are obtained as a weighted combination of the three color channels, as shown in Equation 1:

$$Gr = 0.299R + 0.587G + 0.114B \tag{1}$$





This conversion is directly applied by the $COLOR\_RGB2GRAY$ option of the $cvtColor$ function of the *OpenCV* Python library (Bradski, 2000, https://opencv.org). The spectral response of this grayscale channel is presented in Figure 1 for the OMEA-3C (Figure 1d) and the OMEA-3C-TF (Figure 1e).

The camera sensor and lens are enclosed in a fully anodized aluminum casing system, which is completely watertight (IP67) and protected by a BK7 glass dome in the top. This model features an automatic heating system to defrost the dome and to

avoid rain water droplets on its surface. The control of these cameras is carried out from an independent computer directly connected to them via cable. It is typically housed either in a weatherproof enclosure outdoors or inside a nearby building. The computers of the OMEA cameras belonging to GOA-SCAN are equipped with the *GOA-OMEA-Capture* software, developed by GOA-UVa, to configure settings and manage image capture, downloading, processing, and data transmission (Antuña-Sánchez, 2022).

All OMEA cameras in GOA-SCAN are configured to capture multi-exposure image sequences every 5 minutes during the daytime (with some exceptions set to every 2 minutes) and every 2 minutes at night-time (SZA above 97°). These sequences consist of consecutive image captures taken at different exposure times. For night-time, there are two distinct multi-exposure configurations: Moon, used when the Moon is between the first and third quarters and its zenith angle is below 80°; and Moonless, used the rest of the time. The exposure times for the OMEA-3C camera in Moon mode are 0.1s, 0.5s, 2s, 5s, 10s,

15s, and 20s, while for Moonless conditions, the times are 5s, 10s, 15s, 20s, and 30s. For the OMEA-3C-TF camera, the exposure times differ: 1s, 5s, 10s, 20s, and 30s in Moon mode, and 3s, 6s, 12s, 24s, and 48s in Moonless conditions. The shortest exposures are used to detect the starlight of the brightest stars without saturation, while the longest exposures ensure sufficient signal from the fainter stars. Night-time images are configured to be recorded with an ISO amplification set to a gain of 10 dB ($0.29\ e-/ADU$) for OMEA-3C and 15 dB ($0.16\ e-/ADU$) for OMEA-3C-TF. All images are cropped to

2000x2000 pixels to reduce memory usage while still covering the full sky. The image data are stored in raw format (14-bits in 16-bits format, without demosaicing neither white-balance correction) in a single ".h5" file that includes all images of the same sequence along with additional metadata. *GOA-OMEA-Capture* is programmed to frequently transmit these files to the GOA-SCAN servers, where the data are processed to produce products such as high dynamic range (HDR) images (Antuña-Sánchez, 2022). These HDR images are made available on the GOA-UVa webpage, updated to reflect the latest data from each

GOA-SCAN station (https://goa.uva.es; *Sites and Measurements* tab).

Image data from eight different GOA-SCAN OMEA cameras installed at nine different locations have been utilized in this work. Figure 2 presents a world map highlighting the mentioned stations. As shown, these nine stations cover a wide range of latitudes, including three polar stations: Ny-Ålesund and Andøya (Norway), and the Marambio station (Antarctica, Argentina); as well as two subtropical stations in Canary Islands: Fuencaliente (La Palma, Spain) and Izaña (Tenerife, Spain). Table 1

provides detailed information about these stations and the cameras installed at them, each represented by its ID. All cameras are OMEA-3C, except for the *C013* camera, installed in Valladolid (Spain), which is an OMEA-3C-TF model. Some cameras were relocated, such as the *C005* camera, which was moved from Valladolid to Fuencaliente in September 2021 to monitor the Tajogaite volcanic eruption, and later installed in Izaña and in Andøya. Although some of the mentioned cameras have been



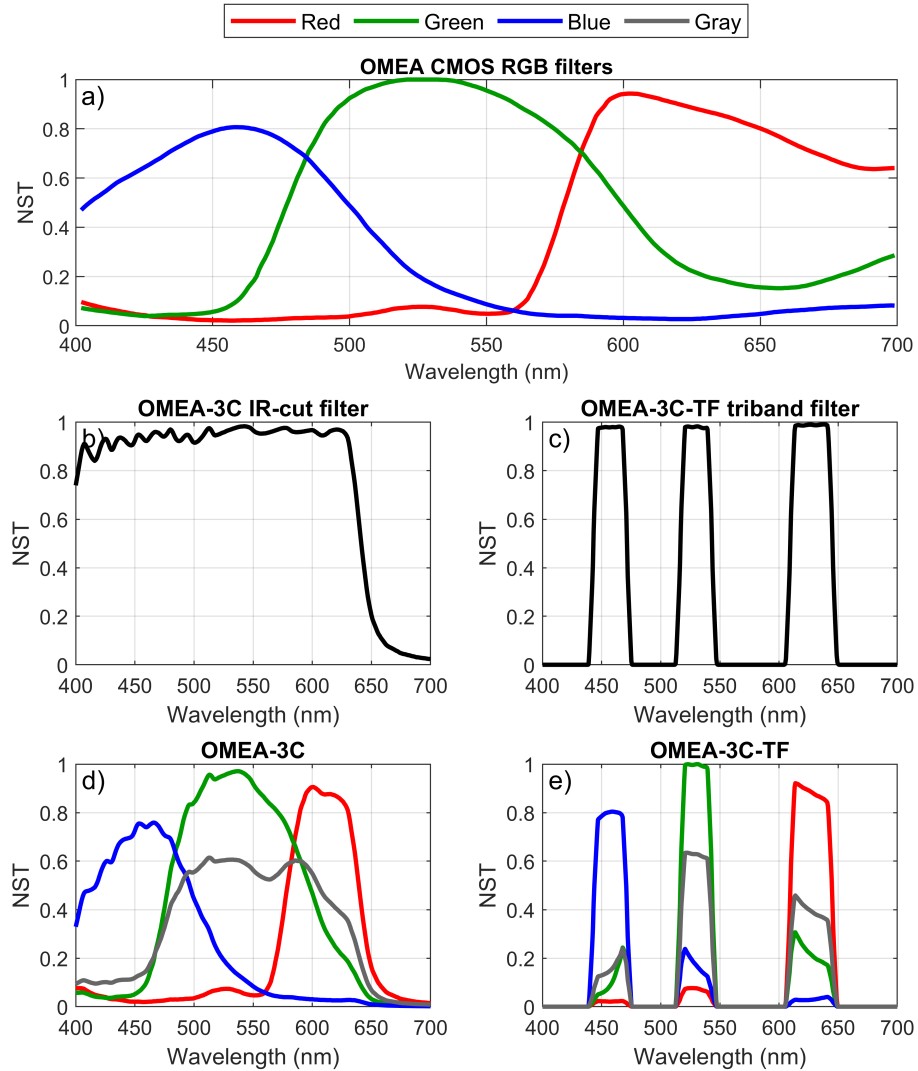

**Figure 1.** Normalized Spectral Transmittance (NST) of different filters: a) Red (R), Green (G), and Blue (B) filters of the Bayer mosaic over the *SONY IMX178* CMOS sensor in the OMEA cameras, normalized to the maximum value among the three channels; b) IR-cut filter included in the OMEA-3C model; c) triband filter included in the OMEA-3C-TF model; d) combination of the RGB filters and the IR-cut filter in the OMEA-3C model: e) combination of the RGB filters and the triband filter in the OMEA-3C-TF model. Panels d) and e) also show the spectral response of the gray signal, which is computed as a weighted combination of the three color channels using the following weights: 0.299 for R, 0.587 for G, and 0.114 for B.

installed since 2018, only data from July 2020 onward have been used in this work, as the *GOA-OMEA-Capture* software was
not available before.





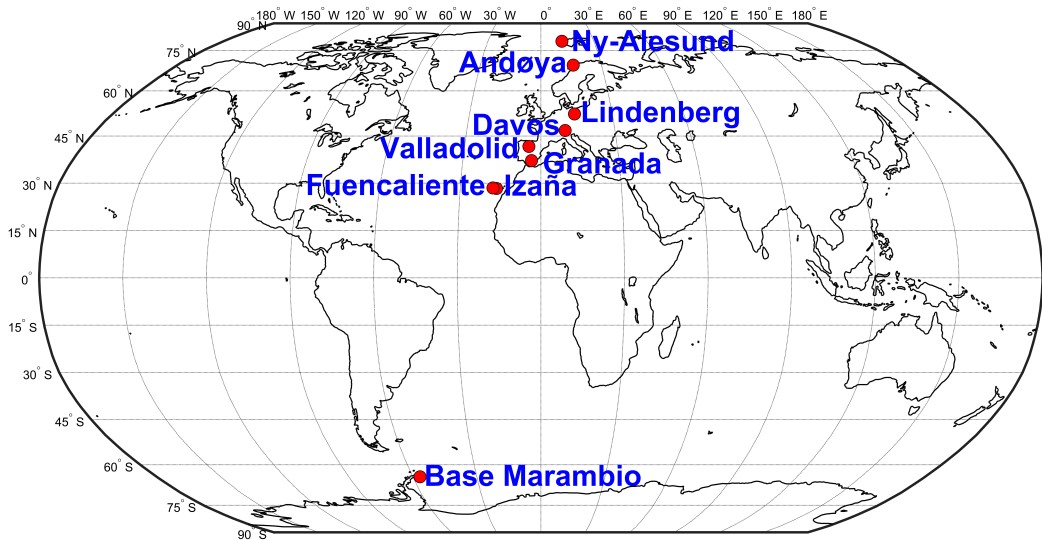

**Figure 2.** Geographical locations of the nine GOA-SCAN stations utilized in this work.

All GOA-SCAN cameras are routinely inspected and cleaned. Each camera, horizontally leveled to point directly at the zenith, undergoes geometric calibration whenever it is relocated or experiences even slight changes in its position. This calibration is carried out using the *ORION* software (Antuña-Sánchez et al., 2022), which determines the azimuth and zenith angles of the celestial vault viewed by each camera pixel, enabling the precise localization of any body within the images if its

coordinates are known.

In addition, each of the stations listed in Figure 2 and Table 1 is equipped with at least one CE318-T sun-sky-moon photometer (manufactured by *Cimel Electronique*). All these photometers are part of AERONET, with periodic calibration, maintenance, and data management overseen by GOA-UVa following the AERONET scheme (Giles et al., 2019). These management tasks are supported by CAELIS (Fuertes et al., 2018, ; https://www.caelis.uva.es), a software tool designed to manage and

process data from photometers controlled by GOA-UVa and to facilitate data visualization. The cloud-screened AOD data collected during daytime (Sun) and night-time (Moon) by these photometers have been directly obtained in this work from CAELIS, with the processing methodology described in detail by González et al. (2020) and Román et al. (2020).

## 3 Methodology

In this section is explained how the starlight signal is extracted from the camera raw data and converted into AOD values.

Stars are punctual sources of light; however, their image in all-sky photos spans several pixels due to long time exposures, star scintillation caused by atmospheric turbulence (Klaus et al., 2004), and the camera's point spread function (Piotrowski et al., 2013). Due to the RGGB Bayer pattern, one-fourth of the pixels record the Blue signal, another one-fourth record the Red signal, and half of the pixels record the Green signal. This means that the star signal is not fully captured by a single color



channel but is distributed across them. Consequently, it is necessary to estimate the signal for each color channel in the remain-
ing pixels of the camera's sensor with different color filters. To achieve this, each 14-bit 2000x2000 raw image is converted
into a 2000x2000x3 RGB image using a demosaicing algorithm: the '$demosaicing\_CFA\_Bayer\_bilinear$' function from
the Python '$colour\_demosaicing$' package (Losson et al., 2010, https://github.com/colour-science/colour-demosaicing). This
method produces three 2000x2000 pixel matrices, one for each color channel, containing the interpolated signal. Additionally,
a fourth grayscale matrix is calculated from the R, G, and B matrices using Equation 1. This grayscale matrix represents a
broader spectral response (see Figure 1), but all its pixels are sensitive to the brightest pixels of the observed stars (this was the
ultimate reason to calculate the Gray channel).

### 3.1 Background

To extract the starlight signal from the obtained images, it is necessary to remove the background signal caused by skyglow,
scattered moonlight, and other factors, such as auroras in polar regions or dark and readout noise frames. The $Background2D$
class, from the $photutils$ (version 1.13.0) Python library (Bradley et al., 2024), has been used for each channel since it is gener-
ally applied to estimate the 2D background noise in astronomical images. $Background2D$ has been configured to use a 10x10
pixels box in which to estimate the background as the median of the pixels within the box ($bkg\_estimator=MedianBackground$).
This method iterates over, each time rejecting values that are less or more than a specified number of standard deviations
($SigmaClip$), which has been chosen as 3 in this work, from a center value. The obtained result has been then smoothed using
a 3x3 median filter ($filter\_size$=(3,3)) to suppress local under or over estimations.

This proposed methodology appears effective for estimating the background in our case, since the stars in the images typ-
ically occupy a small area and exhibit abrupt intensity changes compared to the background. Some parameters, such as the
box size and $SigmaClip$, have been manually selected in this work due to their good performance with the cameras used.
However, the optimal values may vary for other cameras with different characteristics, such as pixel resolution.

Some examples of the background calculation are shown in Figure 3 for different cameras, locations, color channels, and
conditions. The left panels of Figure 3 correspond to the true RGB color composition of each case, after applying a white-
balance correction. Note that although the raw camera signal is recorded in 14 bits, it is stored in 16 bits; consequently, the
digital counts (DC) shown in the colorbars of Figure 3 are scaled to 16 bits.

The upper panels of Figure 3 depict a summer night photo taken at Izaña under the presence of the Moon (Moon phase
angle approximately -40°) and Saharan desert dust conditions (AOD at 440 nm about 0.26). The Gray image (Figure 3b) and
its calculated background (Figure 3c) look very similar; however, after subtracting this background from the original signal
(Figure 3d), the presence of stars becomes evident, indicating that the background correction is apparently not removing the
star signals. Additionally, the differences between the original image and its background reveal that some parts of the horizon,
such as buildings with non-smooth variations, are not perfectly modeled in the background calculation. These horizon parts
can be easily masked to avoid their use. However, Figure 3d also shows that the calculated background underestimates the true
one for pixels close to the Moon, and hot pixels cannot be removed with this method (see some pixels in the corners of Figure



3d), as they exhibit a shape similar to stars. Moreover, some lunar reflections on the camera dome, which appear as circular or linear patterns, are not captured in the background and could be mistaken for stars.

Figures 3e to 3h present a moonless case for the Green channel at Lindenberg, where the Milky Way is accurately estimated in the background (Figure 3g), and a large number of stars become visible when the background is removed (Figure 3h). Another case with the Moon (Moon phase angle of -68°) is shown in Figures 3i to 3l for Valladolid, using the OMEA-3C-TF camera. In this case, a cloud can be observed at the top of the image, which is also captured in the background but underestimated, similar to the Moon and the horizontal city skyline, as shown in Figure 3l. On the other hand, the Orion and Ursa Major constellations are clearly visible in the upper-left and bottom-right parts, respectively, of Figure 3l, where Jupiter is also visible to the left of the Moon. Finally, a case in Davos featuring a red aurora is shown in Figures 3m to 3p. In this case, the background accurately captures the aurora, which is completely removed in Figure 3p. However, a satellite streak can be observed in the left part of Figure 3p, highlighting that line-shaped features are not well captured by the background estimation method used.

## 3.2 Star signal (SS)

Once the background is estimated, it is subtracted from the raw demosaiced image for each channel. This new signal is referred to as the *Background-Corrected Pixel Signal* (BCPS). The light signal of a star should correspond to the sum of the BCPS values of the pixels that capture its light. Therefore, it is necessary to identify these pixels.

To achieve this, the coordinates of the star in the celestial vault for the image must first be determined. This task is performed using the $SkyCoord$ function, based on ephemeris, from the $Astropy$ Python package (version 6.1.1; The-Astropy-Collaboration et al., 2022), taking as input the camera's location coordinates and the image's timestamp, considering the midpoint along the exposure time. If the star's zenith angle exceeds 80°, it is discarded. Then, the closest pixel to the obtained star coordinates is identified using the previously mentioned camera calibration (see Section 2). For each channel, an 80x80 pixel square box centered on this pixel is selected, assuming that the entire star image is contained within this box. An example of this box, centered on the Alnilam star, is shown in Figure 4a for the Red channel of a raw image captured with the C013 camera in Valladolid on January 19th, 2024, at 23:40 UTC, with an exposure time of 20 s (assumed timestamp: 2024-01-19 23:40:45).

The BCPS subimage within the chosen box is expected to contain the image of the selected star, but occasionally other stars may also appear in the same region. This is the case of Figure 4a, where the stars Alnitak and Mintaka also appear in the image, showing the Orion Belt together with Alnilam. A segmentation of this subimage is performed using the $detect\_sources$ function from $photutils$ to identify the stars within the box. This function identifies sources as regions with at least N connected pixels, each exceeding a specified threshold value. In this work, N is set to 10, mainly to avoid hot pixels, and the threshold for each channel is defined as the BCPS median value of all pixels within the box, plus three times the standard deviation of all these pixels. In the example shown in Figure 4, the calculated threshold is 689.6 DC. Figure 4b displays the pixels where the BCPS exceeds this threshold. While the three observed stars, as well as some isolated pixels, surpass this value, only the three stars are identified as such (segments marked in color in Figure 4c) because they have a sufficient number of connected pixels (at least 10 pixels).







**Figure 3.** RGB white-balanced color images for: a) Izaña on July 10$^{th}$, 2022, at 21:16 UTC (camera C005 with an exposure time of 5 s); e) Lindenberg on May 25$^{th}$, 2024, at 00:00 UTC (camera C009 with an exposure time of 30 s); i) Valladolid on January 19$^{th}$, 2024, at 23:40 UTC (camera C013 with an exposure time of 10 s); and m) Davos on May 10$^{th}$, 2024, at 21:30 UTC (camera C038 with an exposure time of 20 s). The second column presents the grayscale raw signal for the Gray (panel b), Green (panel f), Blue (panel j), and Red (panel n) channels, corresponding to the images in the first column. The third column (panels c, g, k, and o) shows the background signal estimated from the respective raw images in the second column. The last column displays the difference between the raw signal (second column) and the estimated background (third column). The colorbars represent pixel digital counts scaled to 16 bits. The maximum values of the colorbars in the second (b, f, j, and n) and third (c, g, k, and o) columns correspond to the 99.5th percentile of the pixel signals in the respective raw images (second column). For the last column, the maximum value is set as the product of the exposure time and 50 digital counts per second.



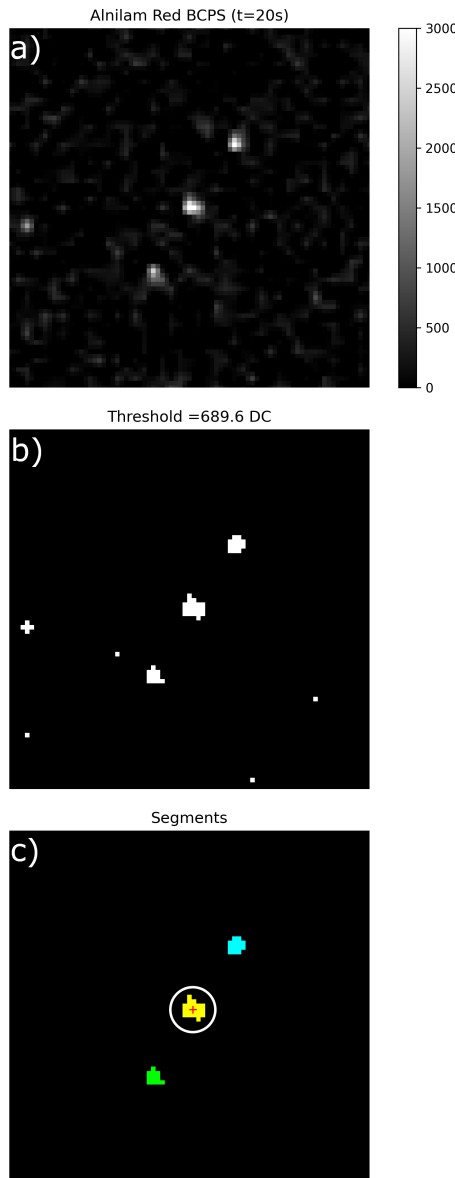

**Figure 4.** a) *Background-Corrected Pixel Signal* (BCPS), in digital counts (DC), subimage within a selected 80x80 pixel box, centered on the Alnilam star, derived from the Red channel of the raw image recorded in Valladolid on January 19[th], 2024, at 23:40 UTC (camera C013 with an exposure time of 20 s). Pixels where the BCPS exceeds the threshold of 689.6 DC are highlighted in white in panel b). Segments identified as stars are marked in color in panel c), where the yellow segment corresponds to Alnilam, the red cross indicates its centroid, and the white circle denotes the region used to extract the star's BCPS.





Additionally, a deblending process is applied using the $deblend\_sources$ function from $photutils$ to separate segments corresponding to overlapping stars, which is a very uncommon occurrence. Subsequently, the centroid coordinates of each segment are calculated using the $SourceCatalog$ function from $photutils$ (see the red cross in Figure 4). The segment with the shortest distance between its centroid coordinates and the analyzed star's coordinates is identified as the corresponding star.

If this distance exceeds 1°, the star is considered as not identified. In the case of Figure 4c, the yellow segment is identified as the Alnilam star.

Once the star is identified in the image, the star signal (SS) can be calculated as the sum of the BCPS values of the segmented pixels identified as star pixels; that is, the sum of the BCPS values of all yellow pixels shown in the example of Figure 4c. However, this method tends to underestimate the actual star signal, as some pixels surrounding the star segment may still

record a fraction of the total signal without exceeding the defined threshold. To address this, a circular aperture with a radius of 5 pixels, centered on the centroid of the star (see the white circle in Figure 4c), is used to encompass the full star signal. Finally, the SS is calculated as the integral of the BCPS within this aperture using the $aperture\_photometry$ function from $photutils$.

### 3.3 Star direct irradiance screening and smoothing

In this work, the SS has been calculated using the described method for all available raw images from the cameras listed in Table 1, and for the following 56 selected stars: Achernar, Acrux, Adhara, Aldebaran, Algol, Alhena, Alioth, Alkaid, Alnair, Alnilam, Alphecca, Alpheratz, Altair, Antares, Arcturus, Atria, Avior, Bellatrix, Betelgeuse, Canopus, Capella, Caph, Castor, Deneb, Diphda, Dubhe, Elnath, Eta Herculis, Fomalhaut, Gacrux, Gamma Herculis, Hadar, Kaus Australis, Kochab, Markab, Merak, Miaplacidus, Mimosa, Mintaka, Mirfak, Mizar, Peacock, Polaris, Pollux, Procyon, Regor, Regulus, Rigel, Rigil Ken-

taurus, Sargas, Shaula, Sheratan, Sirius, Spica, Vega, and Wezen. This list includes the most prominent stars visible from both hemispheres.

Subsequently, the SS of each star is calculated for all the four channels and every available exposure time. However, SS values corresponding to exposure times shorter than 2 s are excluded due to the low signal-to-noise ratio under such conditions. The SS values obtained from at least one saturated pixel are also excluded. In this work, a pixel is considered saturated when its

raw signal (without background correction) exceeds 58,982 DC, a threshold corresponding to 90% of the maximum possible recorded signal in a 16-bit format (65,535 DC). In addition, all SS values obtained for SZA values below 110° are discarded to avoid contamination from sunlight.

The SS values are proportional to the exposure time, representing uncalibrated irradiation (energy per unit area). To standardize the measurements, the obtained SS values are normalized by dividing them by the exposure time, resulting in a metric

proportional to the star's irradiance (power per unit area). This new parameter is referred to as the uncalibrated star irradiance (USI).

The left panels of Figure 5 show the USI values obtained during a winter night at Andøya for various stars. For Procyon (Figure 5a), the signal exhibits a pattern similar to that of direct solar irradiance during daytime under cloud-free conditions, primarily varying with the relative optical air mass ($m$). The recorded $m$ values for Procyon ranged from 5.57 (at star-set and



star-rise) to 2.28 that night, with the USI decreasing as *m* increased. The relative optical air mass in this work is calculated using the next Equation 2 (Kasten and Young, 1989):

$$m = \left( \cos\left(\theta\right) + 0.50572 \cdot \left(96.07995 - \theta\right)^{-1.6364} \right)^{-1} \tag{2}$$

where $\theta$ represents the star zenith angle in degrees.

Stars may rise at the beginning of the night, similar to the Sun during the day, but it is also possible for stars to rise at any

time during the night, or even to set at the beginning of the night and rise again later. The latter case applies to Vega (Figure 5c), which does not even set in the example shown in Figure 5 before starting to rise again when the optical air mass reached 3.19.

Additionally, due to the position of the stars in the sky and the observer's coordinates, the optical air mass of the stars varies to a greater or lesser extent depending on the star. This is illustrated in Figure 5e for Kochab, whose optical air mass ranged

only between 1.25 and 1.00 that night, resulting in very stable USI values throughout the night. An extreme case is Polaris, which is closely aligned with the Earth's North rotational axis, causing its optical air mass to vary only between 1.07 and 1.06 during the night shown in Figure 5g.

In general, the USI values shown in the left panels of Figure 5 appear noisy, with a high deviation, particularly in the Red channel. Noise in the USI values is expected due to shot noise, which is inherent to light, and the uncertainties introduced by the

proposed method for extracting these values. However, Kochab and Polaris, which both exhibit similar USI values (2000–5000 DC/s) and minimal variation in optical air mass, display distinct noise patterns, suggesting an additional noise source.

As noted, the position of Polaris in the sky changes very slightly but is not static; for instance, its azimuth and zenith varied by only 3°and 1°, respectively, during the night shown in Figure 5g. This limited movement ensures that the position of the pixels capturing Polaris does not change rapidly. Consequently, as stars are point light sources, the maximum star signal is

consistently recorded by the same pixel (corresponding to the star's center) for a significant period. This pixel is associated with only one of the three color channels in the Bayer pattern. For instance, if the maximum star signal falls on a Green pixel, it will not be detected by the Blue or Red pixels, and as a result, the demosaiced images for these two channels will not reflect this maximum signal. This explains the behavior of the USI time evolution for Polaris in Figure 5, where all the channels exhibit a cyclic pattern. The maximum of each channel corresponds to the maximum star signal reaching a pixel of that channel,

ensuring that a maximum never appears simultaneously in two different channels. This pronounced oscillation, caused by the use of a single color filter (R, G, or B) per pixel, occurs much faster for all the stars that exhibit more movement across the celestial vault. Consequently, it is also, at least in part, responsible for the high deviation observed in the USI values of these stars.

To reduce this deviation, the USI measurements are smoothed using the following method for each star and channel. For a

target USI value, a time window of ±4 minutes is defined around it. All available USI data within this window are normalized to the optical air mass of the target USI value. This normalization is performed by multiplying each value by its own optical air mass and dividing by the optical air mass of the target USI value. The median of all normalized USI values within the





**Figure 5.** Uncalibrated star signal (USI) at Andøya (camera C005) from January 3 to 4, 2024, for four color channels (Red, Green, Blue, and Gray). The right panels show the corresponding USI values after applying the time-smoothing process. The USI values are presented for the following stars: Procyon (panels a and b), Vega (panels c and d), Kochab (panels e and f), and Polaris (panels g and h).

time window, including the target USI value, is then calculated. Any USI value that deviates by more than 20% (in absolute terms) from this median is considered as outlier and excluded from the time window. A new median is then computed using 305   the remaining normalized USI data, which is taken as the smoothed USI value. Additionally, the standard deviation of the remaining normalized USI values is calculated and used as the uncertainty of the smoothed USI. The number of data points





used in the final median and standard deviation calculations is also recorded ($N_{smooth}$). If $N_{smooth}$ is less than 2, or if the final USI value is not above 1 DC/s, the USI measurement is discarded.

The USI values displayed in the left panels of Figure 5 have been smoothed and are represented in the right panels of the same figure (panels b, d, f, and h). The smoothing process effectively reduces the noise in the data, as can be observed. Additionally, an abrupt decrease in the non-smoothed USI values, caused by the presence of clouds, is noticeable around 16:00 for Vega, Kochab, and Polaris. This decrease is also reflected in the smoothed values; however, some cloud-contaminated data are filtered out due to the presence of outliers and the limited availability of data under these conditions. Consequently, $N_{smooth}$ could serve as a proxy for cloud screening.

Finally, it is remarkable that most of the USI values shown in Figure 5 were derived from images captured under intense auroral activity. However, the effect of these northern lights is not appreciated in the final USI data, highlighting the effectiveness of the proposed background characterization and removal method of Section 3.1. Hereafter, the term USI values will refer exclusively to the smoothed USI values.

### 3.4 Langley calibration and TOD calculation

The direct solar, lunar, or stellar irradiance at one single wavelength reaching the Earth is related to the atmospheric total optical depth (TOD) through Beer-Lambert-Bouguer law. For stars, this relationship can be expressed in terms of the natural logarithm, as shown in Equation 3:

$$\log(USI) = \log\left(USI^0\right) - m \cdot \tau \tag{3}$$

where $m$ is the relative optical air mass, $\tau$ is the TOD, and $USI^0$ is the uncalibrated extraterrestrial star irradiance (i.e., the star irradiance at the top of the atmosphere). The calibration constant that converts USI into well-calibrated physical units is the same constant that converts $USI^0$ into the actual extraterrestrial irradiance. This allows Equation 3 to enable the use of uncalibrated irradiances, thus eliminating the need for tedious laboratory calibrations. If USI is measured for a star, the TOD can then be estimated using Equation 3; only the constant value of $log(USI^0)$ needs to be known for this star.

To this end, the Langley plot calibration method is applied. This method assumes that the TOD remains constant during a period in which the optical air mass varies. Under this assumption, $log(USI^0)$ can be estimated as the y-intercept of a least-squares linear fit of $log(USI)$ as a function of the optical air mass. The slope of the fit corresponds to the total optical depth with a negative sign (see Equation 3).

This method is applied in this study to determine $log(USI^0)$ for each star and channel on a nightly basis. First, rising and setting periods for each star are identified within a single night, as the Langley plot calibration is performed separately for each period. Only $USI$ data recorded under optical air masses between 2 and 5 are used for this purpose, similar to the air mass range employed by AERONET for Langley calibrations (Toledano et al., 2018). The selected $log(USI)$ data are fitted to the optical air mass using a weighted least-squares linear fit, where the weight of each data point is the inverse square of its uncertainty (as determined in Section 3.3). This ensures that data points with higher uncertainty contribute less to the fit.





Once the initial fit is obtained, measured data with an absolute difference greater than 1% compared to the modeled values
from the fit are removed. The weighted least-squares fit is then recalculated using the remaining data. This outlier rejection process is iteratively applied until no additional data points are excluded in a given iteration. For the final iteration, the $log(USI^0)$ (y-intercept) and TOD (negative slope) values and their uncertainty (from the fit) are stored, along with the optical air mass range (maximum $m$ minus minimum $m$), the weighted correlation coefficient ($r_w$), and the number of data points used ($N_{\text{Langley}}$).

From all the obtained $log(USI^0)$ values, only those meeting the following criteria are considered: 1) an optical air mass range greater than 2, to ensure sufficient sensitivity to variations in $m$ during the fit; 2) a weighted correlation coefficient lower than -0.9, to guarantee a strong inverse correlation (the threshold is negative due to the inverse relationship between $log(USI)$ and $m$); 3) $N_{\text{Langley}}$ greater than 40, to ensure an adequate number of data points; and 4) a TOD value (negative slope) below 0.8, to exclude conditions with high aerosol loads.

As example, Figure 6 shows the $log(USI^0)$ satisfying the required criteria for different stars and the four channels of camera C011 during more than three years. These $log(USI^0)$ values exhibit some fluctuations but tend to remain consistent over time for each star and channel. It is noteworthy that this camera was relocated to Granada in 2024, yet no abrupt changes were observed in the obtained $log(USI^0)$ values at this new station. Similar behavior was also observed for camera C005 (not shown), which was installed at various locations. These results suggest that the calibration of the cameras may indeed be
independent of the observation site.

As expected, the $log(USI^0)$ values depend on the star, with higher values observed for brighter stars. This trend can also be seen in Table 2, which presents the median and standard deviation of all available $log(USI^0)$ values for camera C011. Additionally, this table reveals that the Red channel has the lowest availability of $log(USI^0)$ values.

The $log(USI^0)$ value of a star for a given date is calculated as the median of all available $log(USI^0)$ values within a 1-year
time window centered on that date. This approach helps to reduce the uncertainty in the $log(USI^0)$ values. The median is chosen over the mean to minimize the influence of outliers. The uncertainty of these smoothed $log(USI^0)$ values is calculated as the root sum of the squares of two terms: the propagated uncertainty of each individual $log(USI^0)$ value included in the median calculation, and the standard deviation of all averaged $log(USI^0)$ values.

These smoothed $log(USI^0)$ values are also represented as straight lines in Figure 6. Overall, the values remain largely
constant, with minor and gradual time variations, capturing the average behavior of the $log(USI^0)$ data. Caution should be exercised near the edges of the time interval, as the absence of half the averaging period may result in unrealistic values (e.g., the end of the Red channel for Markab in Figure 6a). Such anomalies may resolve themselves as additional data become available in the future. This dynamic method of determining the calibration each day is useful for accounting for possible camera degradations or changes in light intensity from variable stars such as Betelgeuse.

Once $log(USI^0)$ is determined for each available $USI$ measurement, the TOD is calculated using Equation 3. The uncertainty in these TOD values is estimated by propagating the uncertainties of both $log(USI^0)$ and $USI$ through Equation 3.



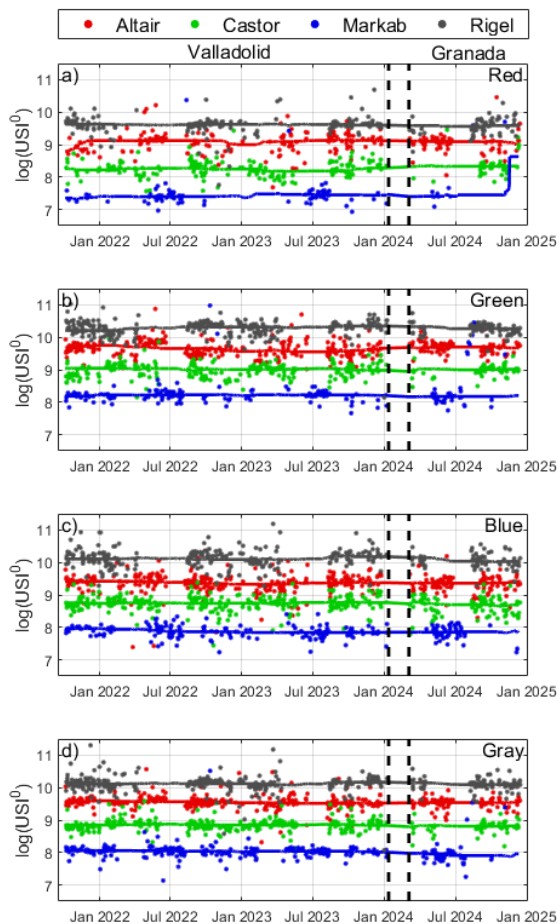

**Figure 6.** Logarithm of the extraterrestrial uncalibrated star irradiance ($log(USI^0)$) obtained using the Langley plot method for camera C011, shown for four stars (Altair, Castor, Markab, and Rigel) and the four color channels: Red (panel a), Green (panel b), Blue (panel c), and Gray (panel d). The straight lines represent the median of $log(USI^0)$ over a 1-year time window centered on the date shown on the x-axis. Dashed vertical lines separate the measurement periods in Valladolid and Granada.

## 3.5 Effective wavelengths

The TOD values are generally associated with an effective wavelength ($\lambda_{eff}$). In photometry, this effective wavelength is usually calculated by considering the spectral response function of the instrument capturing the direct irradiance. In this work, the effective wavelength of each camera channel is determined as the wavelength expected value assuming as a probability



density function the incoming direct spectral star irradiance weighted by the channel's spectral response, as explained by Román et al. (2017) (see Equation 4 in Román et al., 2012):

$$\lambda_{eff} = \frac{\int_\lambda \lambda \cdot S(\lambda) \cdot I(\lambda) \, d\lambda}{\int_\lambda S(\lambda) \cdot I(\lambda) \, d\lambda} \tag{4}$$

where $\lambda$, is the wavelength, $I(\lambda)$ is the direct star irradiance reaching the instrument, and $S(\lambda)$ is the spectral response of the camera channel given by the NST of Figure 1.

To calculate $\lambda_{eff}$, and to ensure realistic values of direct irradiances, the spectral direct transmittance of the Earth's atmosphere is simulated under different conditions using the libRadtran 2.0.4 radiative transfer model (Mayer and Kylling, 2005; Emde et al., 2016). These conditions are the same as those described in Antuña-Sánchez et al. (2021), and include: 7 star zenith angle bins (ranging from 10° to 70° in 10° steps); 5 Ångström Exponent ($\alpha$) values (from 0.2 to 1.8 in steps of 0.4); and 5 Ångström turbidity ($\beta$) values (AOD at 1000 nm, ranging from 0.01 to 0.21 in steps of 0.05). These simulations yield a total of 200 spectral transmittance scenarios.

A realistic incoming star irradiance is obtained by multiplying the extraterrestrial star irradiance spectrum by these transmittance values. However, the extraterrestrial spectrum varies with the star. Therefore, the incoming star irradiance is simulated for 30 different stars, whose extraterrestrial spectra are obtained from the Pulkovo spectrophotometric catalog (Alekseeva et al., 1996). These stars, listed in Table 3, were selected based on their brightness and availability to cover various star types. For each selected star, 200 effective wavelengths (corresponding to the 200 atmospheric conditions) are calculated. The median and standard deviation of these effective wavelengths are presented in Table 3.

Differences in the effective wavelengths among stars are evident, such as Adhara, a blue supergiant with shorter effective wavelengths compared to Gacrux, a red giant. The Green and Gray channels exhibit similar effective wavelengths, although the Gray channel shows a higher deviation due to its broader spectral response. The spectral response differences between the two analyzed camera models (see Section 2) also contribute to variations in effective wavelengths, with longer values observed for the OMEA-3C-TF model, particularly in the Red channel. The standard deviation is comparable between both camera models, except for the Red channel, where it is lower for the OMEA-3C-TF model. The median and standard deviation of the 30 effective wavelengths are calculated and presented in the last row of Table 3 ("All"). These values are assumed to represent the effective wavelengths of the camera channels and, consequently, the wavelengths used for the TOD values. This assumption is made regardless of the star being observed, even though variations between stars are acknowledged. The deviation of the effective wavelength is lower when the bandwidth of the camera channels is reduced with a triband filter like in the OMEA-3C-TF.

## 3.6 Contribution of Rayleigh and gas absorption

To derive AOD, the optical depths (OD) of Rayleigh scattering and gaseous absorption must be subtracted from the TOD. In this work, the gases considered to significantly contribute to the TOD at the camera wavelengths are ozone (O3), nitrogen





dioxide (NO2), and water vapor (H2O). The Rayleigh optical depth (ROD) is assumed to be proportional to ground atmospheric pressure.

The spectral optical depth of each of the aforementioned components can be obtained using Beer–Bouguer–Lambert law. This is achieved by simulating the direct spectral star irradiance at the ground under specific conditions, both with and without the presence of the component, and applying Equation 5:

$$OD_x(\lambda) = m^{-1} \cdot \log\left(\frac{I_{\not{x}}(\lambda)}{I(\lambda)}\right) \tag{5}$$

where $x$ represents the compound for which the optical depth is being calculated, $m$ is the optical air mass, $I(\lambda)$ is the
direct star irradiance at the ground, and $I_{\not{x}}(\lambda)$ is the same $I$ irradiance simulated while neglecting the $x$ compound. The OD values are independent of the selected star. In this work, these spectral OD values are derived using simulations performed with libRadtran 2.0.4 and the REPTRAN parameterization of absorption cross sections of various gases (Gasteiger et al., 2014). The OD is calculated using this method for six different values of each component: total ozone column (TOC) from 200 DU to 400 DU in 40 DU increments; NO2 column from 0.05 DU to 0.425 DU in 0.075 DU increments; total water vapor column
(TWVC) from 2.5 mm to 40 mm in 7.5 mm increments; and the ground atmospheric pressure from 750 hPa to 1000 hPa in 50 hPa increments. When the OD is not explicitly calculated for these parameters, the default values are: 300 DU for TOC, 0.25 DU for the NO2 column, 20 mm for TWVC, and 940 hPa for ground atmospheric pressure.

The calculated OD is valid for extracting the OD at single wavelengths; however, in this case, the camera records the star signal using wider bandwidth filters. Therefore, the OD of each $x$ compound is determined as the expected value of the OD,
using the ground direct star spectral irradiance weighted by the spectral response of the camera as the probability density function (Equation 6):

$$\tau_x = \frac{\int_\lambda OD_x(\lambda) \cdot S(\lambda) \cdot I(\lambda)\, \mathrm{d}\lambda}{\int_\lambda S(\lambda) \cdot I(\lambda)\, \mathrm{d}\lambda} \tag{6}$$

where $\tau_x$ represents the optical depth obtained for the $x$ compound, $OD_x$ is the simulated spectral OD for the $x$ compound, $S$ is the spectral response of the camera channel defined by the NST shown in Figure 1, and $I(\lambda)$ is the simulated ground
direct star irradiance. These spectral $I(\lambda)$ values are calculated for the 30 stars considered in Section 4 (see Table 3), across 7 star zenith angle bins (10° to 70°) and the same 5 bins for the Ångström exponent and turbidity described in Section 3.5. This results in a total of 5250 optical depths for each component and load.

Figure 7 illustrates the calculated optical depth of each compound as a function of its amount for a specific case, and as a function of surface atmospheric pressure in the case of Rayleigh scattering. The obtained optical depths exhibit a linear
behavior. A least-squares fit is performed for each component and channel, revealing high correlation and y-intercept values close to zero, as expected. This indicates that the slope represents an effective absorption coefficient (or a scattering coefficient in the case of ROD), which can be used to derive the optical depth by multiplying it by the total amount of the compound, or by the ground atmospheric pressure in the case of Rayleigh scattering.



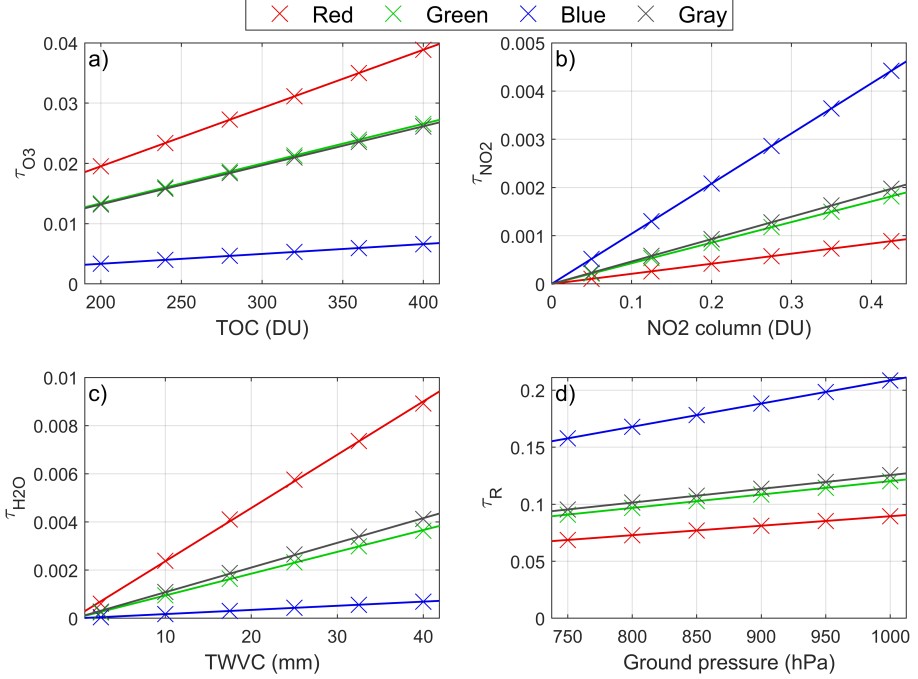

**Figure 7.** Optical depth of ozone ($\tau_{O3}$), nitrogen dioxide ($\tau_{NO2}$), water vapor ($\tau_{H2O}$), and Rayleigh scattering ($\tau_R$) as a function of total ozone column (TOC; panel a), NO2 column (panel b), total water vapor column (TWVC; panel c), and ground atmospheric pressure (panel d), respectively. These values are calculated for the star Regulus with a zenith angle of 50°, and an Ångström exponent and turbidity of 1 and 0.11, respectively. The data are shown separately for the four color channels, with solid lines indicating the least-squares linear fits.

This slope, referred to as the optical depth factor (ODF) in this study, is calculated for the 5250 available cases with varying zenith angles, stars, and aerosol conditions. The mean ODF values for ozone, nitrogen dioxide, water vapor, and Rayleigh scattering are presented in Table 4 for each channel and both camera models. The standard deviation of the ODF values is also included in Table 4, representing the uncertainty associated with the ODF. As expected, the ODF for water vapor is higher in the Red channel, while it is higher in the Blue channel for Rayleigh scattering. The uncertainty on the ODF is lower for the OMEA-3C-TF due to its narrower spectral response.

In this work, the four $\tau$ values are calculated for each available TOD value by multiplying each ODF by its corresponding amount component value. The uncertainty of these retrieved values is also propagated. The hourly TOC and TWVC values are directly obtained from the ERA5 reanalysis (Hersbach et al., 2020), specifically from the dataset ERA5 hourly data on single levels from 1940 to present ($reanalysis-era5-single-levels$; https://cds.climate.copernicus.eu/datasets/reanalysis-era5-single-levels) as the $total\_column\_ozone$ and $total\_column\_water\_vapour$ products.

The ground atmospheric pressure is adjusted to the altitude of each station using an air temperature correction applied to the sea level pressure, with both values obtained from the same ERA5 reanalysis collection: hourly $2m\_temperature$



and $mean\_sea\_level\_pressure$ products. All ERA5 data are retrieved from the Atmosphere Data Store (ADS; https://ads.atmosphere.copernicus.eu/) using the Climate Data Store (CDS) Application Program Interface.

For nitrogen dioxide, data is sourced from the monthly climatological table described by González et al. (2020), derived
from the OMI version 3 dataset (OMNO2d gridded, level 3; Krotkov et al., 2017). All downloaded values (TOC, NO2 column, TWVC and ground pressure) are temporally interpolated to estimate the gas and Rayleigh optical depths, and their uncertainties, at the times corresponding to the available TOD values.

### 3.7   AOD and cloud-screening

Once the TOD and its corresponding gas and Rayleigh optical depths are available, the AOD is calculated for each star as the
TOD minus each of the four mentioned optical depths (O3, NO2, H2O, and Rayleigh). These AOD values are represented in the left panels of Figure 8 for several stars and the four effective wavelengths during one night at Izaña.

The AOD is available for 37 different stars that night, but only 7 randomly chosen stars are shown to improve visualization in the plots; error bars are omitted for the same reason. The AOD values generally overlap in time for different stars. However, the AOD occasionally shows significant deviations for some stars compared to others. For instance, this is well observed in
Figure 8 in the AOD from Mirfak in the middle of the night, especially at 591 nm (Red channel) and 534 nm (Green channel). These deviations of the AOD from other stars can at least partially be attributed to inaccuracies in the $log(USI^0)$ calibration constant. Additionally, the presence of clouds in the star's position can also cause AOD differences between stars; however, such effects should be noticeable across all channels.

Moreover, the AOD for some stars exhibits pronounced oscillations, such as Mirfak in the middle of the night or Pollux
at the end. These oscillations primarily occur under conditions of low $N_{smooth}$ values and low optical air mass. This suggests that the oscillations may be caused by the insufficient number of USI values in the smoothing process described in Section 3.3. The lack of data could be partially related to low optical air masses, where the field of view of these pixels is larger due to the fisheye lens projection. This causes starlight to be concentrated in fewer pixels at low zenith angles, leading to the star segment not occupying enough pixels to be considered. Additionally, the higher concentration of starlight makes the signal in
these pixels more prone to saturation under these low optical air mass conditions.

A cloud-screening and quality assurance (CS/QA) method is proposed to filter the AOD data series for each star and channel to exclude cloud-contaminated, inaccurate, and spurious data. First, any AOD value whose sum with its uncertainty falls below zero is removed, as it lacks physical meaning. Additionally, only AOD data with $N_{smooth} > 10$ are considered. To reduce temporal fluctuations, a running mean filter, similar to the proposed by Pérez-Ramírez et al. (2012), is applied: any AOD value
showing an absolute difference greater than 0.02 from the mean AOD calculated within a $\pm5$-minute window (avoiding the self AOD value) is discarded. If no other AOD data exist within the time window, the value is also rejected. This running mean method is applied iteratively until no further data are removed. Finally, to eliminate outliers and spurious data, a $3\sigma$ criterion, similar to the described by Giles et al. (2019), is employed: the median and standard deviation of the remaining AOD values are calculated, and any AOD value falling outside the range defined by the median $\pm3\sigma$ (being $\sigma$ the standard deviation) is
rejected. This process is repeated iteratively until no further data are discarded or the standard deviation falls below 0.02.



**Figure 8.** Aerosol optical depth (AOD) at Izaña (camera C005) from November 3 to 4, 2022, at different wavelengths for the following stars: Adhara, Alhena, Dubhe, Mirfak, Pollux, Regulus, and Vega. The right panels show the corresponding AOD values after filtering for clouds and spurious data. The AOD values are presented for the following wavelengths: 591 nm ($AOD_{591}$; panels a and b), 534 nm ($AOD_{534}$; panels c and d), 466 nm ($AOD_{466}$; panels e and f), and 537 nm ($AOD_{537}$; panels g and h).

The right panels of Figure 8 present the same AOD values as those in the left panels, but filtered using the method explained above. As observed, the filtering method successfully removes AOD values with higher deviations and fluctuations, leaving most of the remaining AOD values close to low values. The data reduction is more pronounced in the Red channel. These



filtered data are shown for only seven stars, but filtered AOD values from a total of 36 different stars are available for that
night, one fewer than the unfiltered data series. This indicates that the AOD data for one star is entirely removed after filtering.
That star is Sirius, which is one of the brightest stars and frequently appears saturated, resulting in $N_{\text{smooth}}$ values that are
consistently below the required threshold.

The differences in the filtered AOD obtained from different stars are likely to be primarily due to calibration characteristics, although part of these differences could also arise from inhomogeneities in the aerosol spatial distribution and certain
assumptions, such as using the same effective wavelengths and gas and Rayleigh optical depths for all stars. Assuming spatial
homogeneity, the AOD values from different stars are averaged to produce a single AOD value per channel at each time. This
averaging process is explained next.

First, for each channel, the filtered AOD data from all stars are grouped into 5-minute intervals centered on specific times:
on the hour, 05 minutes, 10 minutes, and so on. For example, one group is formed with data in the interval 19:57:30 < time
$\leq$ 20:02:30, which corresponds to 20:00:00; the next interval is 20:02:30 < time $\leq$ 20:07:30, assigned to 20:05:00, and so
forth. Then, a $3\sigma$ criterion is applied to each AOD group to remove outliers in a similar way than Giles et al. (2019): the AOD
median and standard deviation are calculated, and any data point outside the interval median$\pm3\sigma$ is removed. This process
is iteratively repeated until no further data points are removed or the standard deviation falls below 0.02. Once outliers are
rejected, the weighted mean of the remaining data is calculated, with the weight of each data point being the inverse square of
its estimated uncertainty. This weighted mean is then assumed to represent the AOD for the analyzed channel and time interval.
The uncertainty on this AOD is calculated by propagating the uncertainty of the averaged AOD data. The standard deviation
of the averaged data ($\sigma_{5min}$), the number of averaged data ($N_{5min}$), and the number of different stars involved in the averaged
data ($N_{star-5min}$) are stored for quality control purposes.

An AOD value is then obtained for each channel every 5 minutes. Figure 9 presents these averaged AOD values corresponding to the case depicted in Figure 8. The AOD remained relatively stable throughout the night, with values around 0.04 across
all channels. As expected, the Red channel generally exhibits lower AOD values compared to the other channels; however, it
also shows larger temporal fluctuations and higher uncertainties.

These averaged data, which are analogous to AERONET level 1.0 data, can still be contaminated by clouds or may lack
sufficient confidence. Therefore, an additional CS/QA method is applied to filter these data. First, for quality assurance, AOD
values below 0 are rejected due to their nonphysical nature. Additionally, all AOD data with $N_{5min} \leq 10$, $N_{star-5min}<2$ or
$\sigma_{5min} \geq 0.08$ are removed, as they indicate an insufficient number of observations or a high level of deviation.

For cloud-screening, a methodology similar to AERONET version 3 is applied (Giles et al., 2019). This process considers
all the remaining AOD data from a full night and is applied independently to each channel. The procedure consists of the
following steps:

1. Smoothness Criterion: If two consecutive AOD points show an AOD difference greater than 0.01 per minute, the higher
AOD value of the two is removed. This step is applied iteratively until no further data is removed.

2. Standalone Criterion: If no other AOD data remains within a $\pm1$-hour time window around a given AOD data point, that
data point is removed.



3. $3\sigma$ Criterion: The standard deviation ($\sigma$) of all remaining AOD data in the night is calculated. If $\sigma > 0.02$, any AOD values falling outside the interval defined by the median $\pm 3\sigma$ are discarded. This step is applied only once (no iterative).

4. Repeat Standalone Criterion: The standalone criterion is reapplied to the remaining data.

5. Minimum Data Requirement: If the number of remaining AOD values is below 3 or less than 10% of the initial available averaged values, all the data for that night are removed.

This methodology ensures that only high-quality AOD data remains after the screening process, being analogous to AERONET level 1.5 data.

This CS/QA method is applied to all available data. In the case of the AOD shown in Figure 9, the data passing this CS/QA method is displayed in blue, while the removed data is shown in red. Green and Gray channels do not exhibit any removed data, and only one data point is removed for the Blue channel due to a low $N_{5min}$ value of 7. However, the Red channel presents several data points removed by the quality assurance criteria of the CS/QA method: some points at the beginning and end of the night exhibit AOD values below zero; the data removed at the beginning of the night, although above zero, and the five consecutive points around 20:20 have $N_{5min}$ values $\leq 10$, consistent with the observed in Figure 8b; and the data points at 04:45 and 04:55 are removed because $N_{star-5min}=1$ and $\sigma_{5min}=0.09$, respectively.

## 4   Results

The methodology explained in the previous section has been applied to all available camera data (Table 1), yielding cloud-screened and QA-filtered AOD values at four effective wavelengths corresponding to the Red, Green, Blue, and Gray camera channels. To evaluate the performance of the proposed methodology, the retrieved AOD values are compared against independent AOD measurements taken at night-time at the same stations using a CE318-T sun-sky-moon photometer.

### 4.1   AOD comparison with moon photometer data

The AOD data from the photometer, obtained through CAELIS, do not correspond to the same wavelengths as those retrieved from the camera. To match them, the photometer AOD values at 440 nm, 500 nm, and 675 nm are selected, as these are the closest to the camera wavelengths. Using these three AOD values, the Ångström Exponent and the turbidity parameter are calculated. The photometer AOD is then interpolated to the camera wavelengths using these two Ångström parameters.

To temporally match the two datasets (camera and photometer), for each AOD data point from the camera, photometer data within a $\pm 2.5$-minute window for the same wavelength are selected. These data within this window are averaged, and this mean value is assumed as the reference for comparison with the camera AOD.

Once the two datasets are matched in time and wavelength, they are directly compared for all the different cameras and wavelengths in Figure 10. The color scale in this figure indicates the number of camera-photometer data pairs found within AOD pixels of size 0.02x0.02.

In general, the AOD from the cameras shows values similar to those from the photometer, with most of the data points lying close to the 1:1 line and exhibiting correlation coefficient values above 0.90 in most cases. However, this correlation

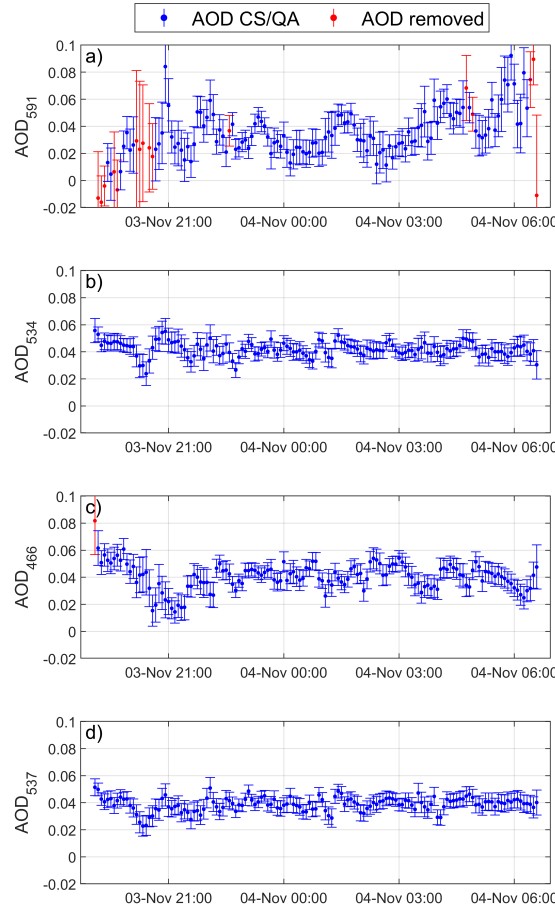

**Figure 9.** Aerosol optical depth (AOD) at Izaña (camera C005) from November 3 to 4, 2022, at 591 nm ($AOD_{591}$; panel a), 534 nm ($AOD_{534}$; panel b), 466 nm ($AOD_{466}$; panel c), and 537 nm ($AOD_{537}$; panel d). Error bars indicate $\pm$ the AOD uncertainty. AOD values that do not meet the cloud-screening and quality assurance (CS/QA) criteria are shown in red and are removed from the final data series represented in blue.

is significantly lower for cameras C003 (Marambio) and C004 (Ny-Ålesund), especially in Marambio, where a considerable number of outliers can be observed. These outliers may be due to camera data contaminated by clouds or failing to meet quality criteria that were not properly filtered by the developed algorithms. They may also result from photometer data not properly filtered for clouds, as thin clouds are harder to detect at night-time without lunar aureole measurements. This issue is evident in panels a), b), c), and d) of Figure 10, where high photometer values appear in the bottom-right corner for low camera AOD






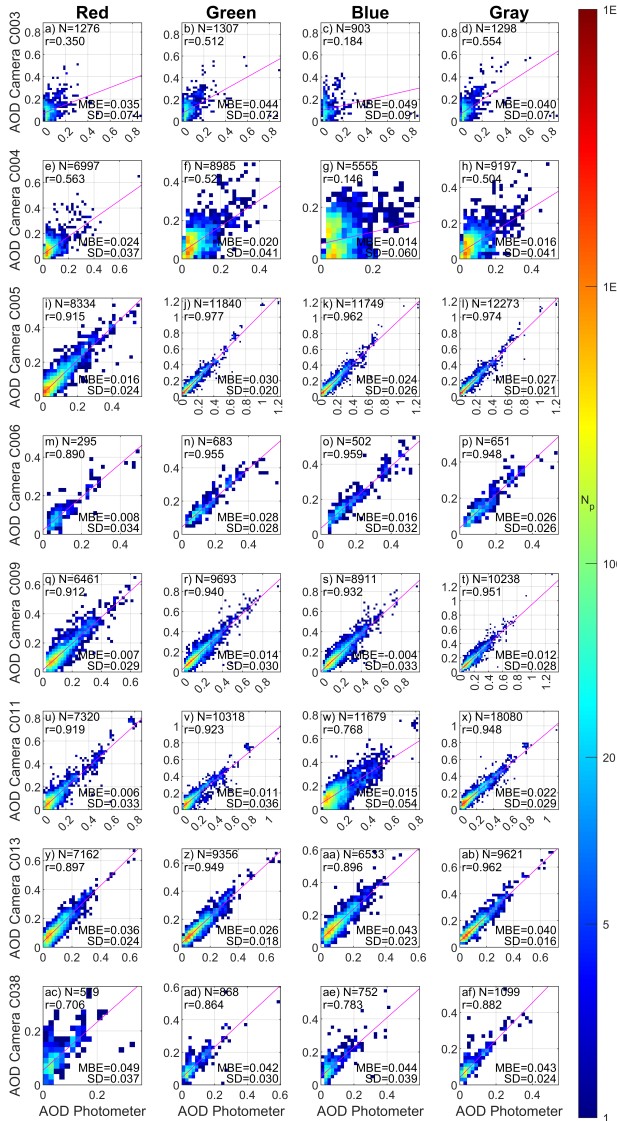

**Figure 10.** Density plots representing the AOD values retrieved from the cameras against those obtained by CAELIS from the sun-sky-moon photometer measurements at night-time. Each row corresponds to a specific camera, and the columns represent the Red, Green, Blue, and Gray channels, respectively. The Red, Green, Blue, and Gray channels correspond to effective wavelengths of 614 nm, 538 nm, 472 nm and 548 nm, respectively, for the C013 camera; 591 nm, 534 nm, 466 nm and 537 nm for the remaining cameras. The density is represented by the colorbar, which indicates the number of available data points ($N_p$) in a square bin with a pixel size of 0.02 AOD units (2D histogram). The number of available data points (N), the correlation coefficient (r), the mean bias error (MBE), and the standard deviation of the differences between both AODs (SD) are included in each panel. The solid magenta line represents the least-squares linear fit.





values. Although this problem with outliers is common across all cameras (e.g., panel i)), it is more pronounced in Marambio due to the high cloud presence and generally low AOD values at this station.

In the case of the camera C004 in Ny-Ålesund, the differences between the camera and the photometer may partly be due to the Langley calibration method, which is ideal for low-latitude regions but performs worse at high latitudes due to slower
changes in the optical air mass. Additionally, the calibration values $log(USI^0)$ are obtained as the annual average of $log(USI^0)$ values that meet certain requirements based on static thresholds, which may not be optimal for all stations.

For a more detailed quantification of the differences between both datasets, Table 5 presents several statistical indices comparing the two datasets, using the photometer as the reference. Generally, fewer AOD data points are available from the camera for the Blue and especially the Red channels, a pattern also observed in Figure 9. The correlation coefficient is also generally
lower for these channels. Table 5 presents the mean (Mean Bias Error; MBE), median (Md), and standard deviation (SD) of the differences ($\Delta$) between the AOD from the cameras and the photometer. The MBE, which represents the accuracy of the camera AOD, ranges between 0 and 0.05, with the highest values observed for cameras C038 (Davos) and C003 (Marambio), and the lowest values for cameras C011 and C009 (Lindenberg). In general, the camera AOD overestimates the photometer AOD for all cameras, with this overestimation being approximately about 0.02 for all wavelengths when considering data from
all cameras.

The SD values, which represent precisionranges from 0.02 to 0.06, except for the camera C003 at Marambio, which shows values of 0.09 for Blue channel, likely due to the aforementioned outliers. The lowest SD values are found for camera C013, which is the only one with a triband filter, followed by C005. The Blue channel generally shows the worst precision, with higher SD values of approximately 0.05 when considering all cameras, while this SD is close to 0.03 for the other wavelengths.
The slope ($b$) and y-intercept ($y_0$) of the linear fit shown in Figure 10 are also presented in Table 5. The $y_0$ values are low but positive, and most $b$ values are below 1, indicating a linear fit that tends to overestimate low AOD values while underestimating high ones.

Table 5 also shows the percentage of the absolute values of $\Delta$ that fall below the standard uncertainty ($\Delta < \sigma$) and the expanded uncertainty ($\Delta < 2\sigma$). The standard uncertainty is defined as the sum of the errors of the camera and photometer
AODs, while the expanded uncertainty is twice the standard uncertainty. In this case, the uncertainty associated with the photometer AOD is assumed to be 0.02. If the uncertainty associated with the camera AOD properly represents its error as a Gaussian probability density function with the standard deviation equal to the uncertainty, then the values of $\Delta < \sigma$ and $\Delta < 2\sigma$ should approximate 68% and 95%, respectively. The values of $\Delta < \sigma$ vary depending on the camera and wavelength, as shown in Table 5. However, in most cases, except for camera C003, the $\Delta < 2\sigma$ values are close to the mentioned 95%. In
fact, when considering data from all cameras combined, the $\Delta < 2\sigma$ values range between 93.6% and 97.3%, indicating that the estimated AOD uncertainty accurately reproduces the real uncertainty in the camera AOD.

Cameras C005 and C011 were installed in various locations; however, the results in Table 5 do not account for this detail. To evaluate the performance of these cameras in each location, Table 6 presents the same statistical indices as Table 5, but distinguishing between the locations for cameras C005 and C011.





The $r$ values are generally lower for Andøya compared to the other locations of camera C005. Similarly, for camera C011, r values for Valladolid are lower than those for Granada. Differences between locations also appear in the MBE, with the most significant difference observed in the Blue channel of camera C011, where the MBE in Granada reaches the lowest value across all cameras (-0.01). SD values for Izaña are the lowest among all cameras and locations. This improved precision may be related to Izaña being an ideal site for Langley calibrations due to its low latitude and high altitude, which provide a clean and

stable atmosphere (Toledano et al., 2018). On the other hand, the SD values in Andøya are higher than in other locations, likely because Andøya is a less suitable site for Langley calibration due to its higher latitude. This variation in SD between locations observed for camera C005 contrasts with the similar SD values obtained at the two locations (Valladolid and Granada) for camera C011. The $\Delta < 2\sigma$ values remain close to the expected 95% for all locations.

Therefore, the performance of the cameras in estimating AOD appears to depend on the location where they are installed.

On the other hand, the differences in the results from Table 5 between cameras C006 and C009, both located in Lindenberg, are not as pronounced and may be caused by the insufficient number of data available for C006. Overall, it seems that the main differences observed between the various cameras are attributable to the location where they are installed rather than the camera itself, at least when considering OMEA-3C cameras (all except C013).

## 4.2    Case study

In the previous comparison (Section 4.1), the agreement between the AOD from the cameras and the photometer at night-time is discussed. However, the photometer AOD at night-time is only available when the Moon is visible and between the first and last quarters (less than half of the time), while the camera AOD is available throughout the entire night since stars are always visible in the sky. To evaluate the goodness of the AOD obtained from the cameras during moonless night-time, Figure 11 shows the time series of the camera AOD over 20 days in Izaña, along with the daytime AOD obtained with the photometer

to assess whether the day-to-night transition in the AOD is consistent. During this period, the night-time AOD values from the photometer, when the Moon was visible, are also included.

During the period shown, it can be observed how the daytime AOD varied throughout the days, ranging from values close to zero to values above 0.4. The increases in AOD above the typical low values for this station were caused by the arrival of Saharan dust from North Africa. As can be observed for all wavelengths, the night-time AOD evolution aligns well with the

daytime values, even when Moon data is not available. For instance, the increase in daytime AOD from the end of 7 July to the early morning of 8 July is well captured by the camera AOD, which shows a clear increase starting just after midnight. A similar behavior is observed between 21 and 22 July. In general, AOD from camera correlates with the expected values even under moonless conditions.

One of the poorest performances of the camera AODs against the photometer occurs in Ny-Ålesund (C004), where half of

the year is continuously dark. As an example, Figure 12 shows the AOD over five consecutive days in December 2023 at this location. First, it can be observed that there are no daytime data, while night-time data are continuously available due to the time of year. On December 22, 23, and 24, AOD values around 0.1 are observed for all wavelengths, showing a smooth and fairly constant temporal evolution. A similar trend can be seen in the photometer data. Although the camera AOD shows some



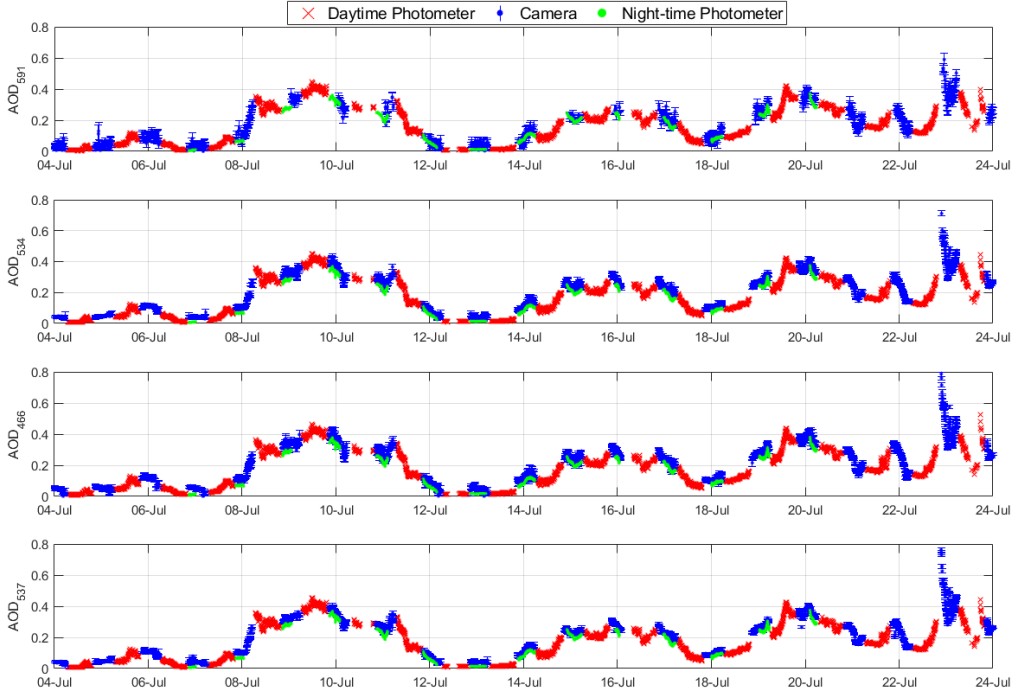

**Figure 11.** AOD at 591 nm (AOD$_{591}$), 534 nm (AOD$_{534}$), 466 nm (AOD$_{466}$), and 537 nm (AOD$_{537}$) at Izaña for 20 days in July 2022. Blue error bars represent the night-time camera AOD $\pm$ its associated error. The red crosses and green dots correspond to the AOD data from the sun-sky-moon photometer during daytime and night-time, respectively.

differences compared to the reference AOD obtained from the photometer, it is still useful for observing the temporal evolution
and magnitude of the AOD. Notably, the camera AOD at 466 nm shows lower values compared to the photometer and the other wavelengths. This could be caused by inaccuracies in the Langley calibrations, which, as mentioned above, are less accurate when conducted at high latitudes.

The data from the Lidar (not shown) installed in Ny-Ålesund reveals the presence of polar stratospheric clouds (PSCs) between 21 and 23 km altitude from 07:00 UTC on the 23 to 19:00 UTC on the 24 December. These clouds have a very low
optical depth, which prevents the cloud-screening algorithm from filtering out AOD data in their presence. Therefore, part of the AOD observed in Figure 12 is influenced by these PSCs. These clouds are also not filtered out by the photometer, as high clouds are typically screened during the day using solar aureole radiance measurements (Giles et al., 2019). However, at night, lunar aureole measurements are not available for this cloud screening. Additionally, Lidar data indicate that the last AOD measurements on the 25 December are likely contaminated by high ice clouds, such as cirrus, which are known to be
more challenging to filter. The same issue may also affect the first AOD measurements on the 21 December, although no Lidar



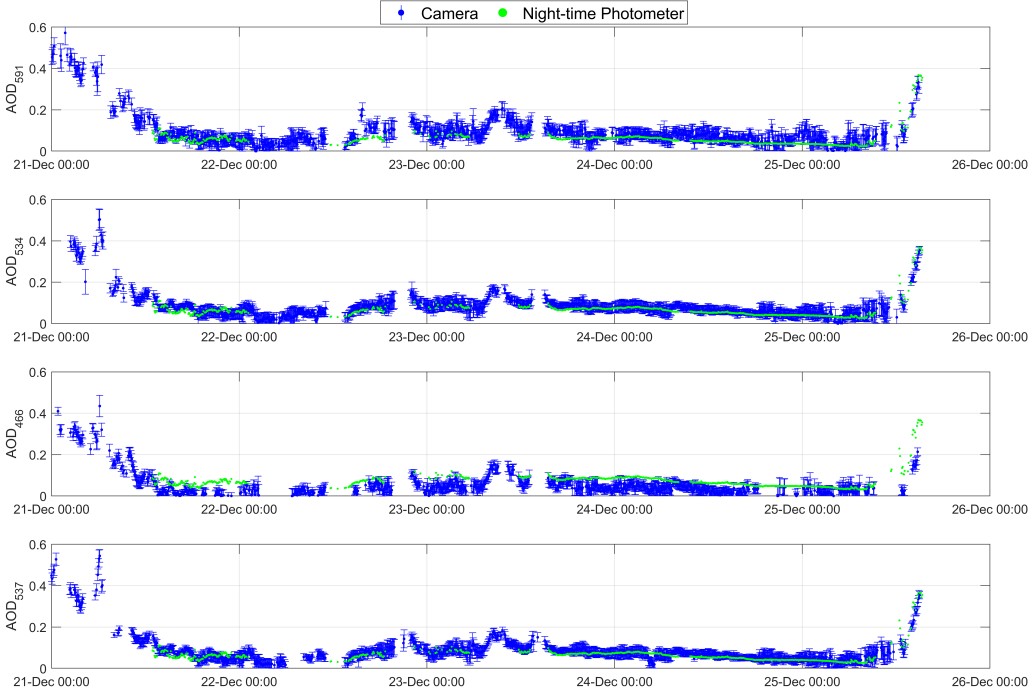

**Figure 12.** AOD at 591 nm (AOD$_{591}$), 534 nm (AOD$_{534}$), 466 nm (AOD$_{466}$), and 537 nm (AOD$_{537}$) at Ny-Ålesund for 5 days in December 2023. Blue error bars represent the night-time camera AOD $\pm$ its associated error. The green dots correspond to the AOD data from the sun-sky-moon photometer during night-time.

data are available to confirm this. These findings highlight that conditions during the polar night can indeed be variable, and cloud-screening algorithms could be further improved to better detect high and thin clouds.

## 5  Conclusions

This study has demonstrated the potential of using all-sky cameras to estimate atmospheric properties such as the total optical

depth (TOD) and the aerosol optical depth (AOD) under night-time conditions. This paper has proposed a novel method for extracting the starlight reaching the Earth for different stars from night-time sky images captured by an all-sky camera. This recorded starlight is equivalent to an uncalibrated irradiance, and then it can be used for star photometry transforming this starlight into the AOD.

Langley calibration has been used to calculate the TOD and, after properly accounting for the effects of gas scattering

and absorption, to retrieve the AOD at four different effective wavelengths (Red, Green, Blue and Gray camera channels). A comparison against independent and accurate photometer measurements at nine different locations, has revealed that the



AOD values retrieved from the cameras correlate with the photometers ones, with correlation coefficients exceeding 0.9 in many cases. Additionally, when considering data from all stations combined, the mean bias error (accuracy) and the standard deviation (precision) of the camera-photometer differences are approximately 0.02 and between 0.03 and 0.04, respectively.

However, the worst agreement appears for higher latitude locations, likely related to the limitations of Langley calibrations in such regions.This suggests that alternative methods or cross-calibration with other stations may be necessary for such areas. Additionally, the Langley calibration is calculated as an average over a $\pm 6$-month window, meaning it is updated when new data becomes available after at least six months. This highlights the importance of exercising extra caution when interpreting data from the most recent six months.

The proposed methodology to extract starlight, identify valid Langleys, filter cloudy and low-quality data, etc., is based on a series of approaches that use several threshold values manually established. The method and thresholds are assumed to be the same and independent of the measurement station. However, these thresholds may need to be adjusted differently depending on the measurement station for optimal performance. Additionally, the list of stars could be refined and tailored to each specific location. Moreover, for all the measurements an all-sky image is available, where the presence of clouds can be observed 665    globally, and it could be used for cloud-screening in future.

The use of RGB cameras allows for obtaining starlight in different spectral bands and, therefore, at various effective wavelengths. However, since stars are point sources and the camera sensor has a mosaic of color filters, such as the RGGB Bayer pattern, this influences the recorded starlight signal. Additionally, the standard camera color filters are usually spectrally wide, which introduces greater inaccuracies in photometry purposes, such as calculating an effective gaseous absorption. Reducing 670    this spectral width improves the precision of the AOD, as observed when comparing the performance of different camera models in Valladolid. With all this in mind, an improvement to the method could involve using cameras without a filter mosaic and with narrower spectral responses, such as incorporating an interference filter like those used in photometers. While this would address the highlighted issues, it is important to consider that the amount of starlight captured by the camera would be significantly lower. This would require longer exposure times and/or higher amplification gains, leading to fewer images and/or 675    an increase in noise.

Although the use of a camera with a color filter mosaic has enabled the calculation of AOD at different effective wavelengths, the small spectral range (less than 150 nm) and the inherent uncertainty in the calculated AOD do not allow for the reliable estimation of the Ångström Exponent.

Even with these limitations, the proposed methodology provides AOD values similar to actual ones, serving at least as a good 680    proxy. This method has proven useful for detecting and monitoring temporal changes in AOD, such as those caused by Saharan dust intrusions, highlighting its potential for studying aerosol dynamics. This, combined with the ability to continuously monitor AOD throughout the night, regardless of moonlight, offers an unprecedented opportunity to fill the gap in night-time AOD measurements without being restricted to lunar visibility.

Future research should focus on improving data filtering algorithms and calibration methods for high-latitude stations, testing 685    other camera models, and further validating the proposed methodology under different atmospheric conditions. We encourage scientific researchers involved in atmospheric studies to use the data generated from this work and to adopt the proposed



methodology (either by joining GOA-SCAN or implementing it themselves) to calculate AOD with their own all-sky cameras. Additionally, we invite other researchers to compare the methodology presented in this work with measurements performed in other locations and to work on improving it, so the entire community can benefit from advancements in this promising research

line. Finally, we recommend to all-sky camera manufacturers considering these new proposed ideas to adapt their technical developments to the specific needs of this new application.

*Author contributions.* Conceptualization, R.R.; methodology, R.R., D.G. and J.A.; software, R.R., D.G. and J.A.; validation, R.R, C.H. and R.G.; formal analysis, R.R. and D.G.; investigation, R.R., D.G. and V.C.; data curation, A.B., L.D., C.R., N.K., G.C., and M.G.; writing— original draft preparation, R.R. and D.G.; writing—review and editing S.H., R.G., C.R., A.B., V.C. and C.H.; supervision, D.M, and C.T.;

project administration, R.R., V.C., C.T., A.C. and D.M.; funding acquisition, C.T., V.C. and A.F. All authors have read and agreed to the published version of the manuscript.

*Competing interests.* No competing interests are present.

*Acknowledgements.* This work is part of the project TED2021-131211B-I00375 funded by MCIN/AEI/10.13039/501100011033 and European Union, "NextGenerationEU"/PRTR. This work was supported by the Ministerio de Ciencia e Innovacion (MICINN), with the

grant no. PID2021-127588OB-I00 and is based on work from COST Action CA21119 HARMONIA. Financial support of the Department of Education, Junta de Castilla y León, and FEDER Funds is gratefully acknowledged (Reference: CLU-2023-1-05). The authors acknowledge the support of the Spanish Ministry for Science and Innovation to ACTRIS ERIC. Grant PID2022-142708NA-I00 funded by MCIN/AEI/10.13039/501100011033 is also acknowledged. We want to thank to Rogelio Carracedo, Javier Gatón, José Luis Martín-Marcos, Patricia Martín-Sánchez (GOA-UVa staff), for the maintenance of the instrumentation and support for the station infrastructure at Valladolid.

This acknowledgement is also extended to the staff from AEMet, MOL-RAO (DWD), AWIPEV (specially Sandra Graßl), PMOD-WRC, GFAT and SMN in charge of the maintenance and cleaning of the all-sky cameras at Izaña, Fuencaliente, Lindenberg, Ny-Ålesund, Davos, Granada, and Marambio. We thank to the FMI (Finnish Meteorological Institute) for the help with the instrumentation deployment in Marambio, and to Ricardo Sánchez for his help with data management.



## Acronyms

| | |
|---|---|
| AEMet | Agencia Estatal de Meteorología: Meteorological State Agency of Spain |
| AERONET | AErosol RObotic NETwork |
| AOD | Aerosol Optical Depth |
| ASC | Andøya Space Center, Andøya, Norway |
| AWI | Alfred-Wegener-Institute: Helmholtz Centre for Polar and Marine Research (German Polar Institute) |
| AWIPEV | Alfred Wegener Institute Paul Emile Victor: AWI and IPEV German-French Arctic Research Station (aka "Koldewey-Station") in Ny-Å, Svalbard |
| BCPS | Background-Corrected Pixel Signal |
| CS | Cloud screening |
| DC | Digital Counts |
| DU | Dobson Units |
| DWD | Deutscher Wetterdienst: German Meteorological Service |
| GFAT | Grupo de Física de la Atmósfera de la Universidad de Granada (UGR): Atmospheric Physics Group of the University of Granada (UGR), Granada, Spain |
| GOA | Grupo de Óptica Atmosférica: Atmospheric Optics Group of UVa, Valladolid, Spain |
| GOA-SCAN | GOA all-Sky CAmeras Network |
| Gr | Grayscale |
| GRASP | Generalized Retrieval of Atmosphere and Surface Properties, Lille, France |
| HDR | High Dynamic Range |
| IARC | Izaña Atmospheric Research Center: AEMET Observatory of Izaña, Tenerife, Spain |
| IISTA-CEAMA | Instituto Interuniversitario de Investigación del Sistema Tierra - Centro Andaluz de Medio Ambiente: Andalusian Institute for Earth System Research - Andalusian Center for the Environment, UGR, Granada, Spain |
| IPEV | Institut polaire français Paul Émile Victor: French Polar Institute |
| IR | Infrared |
| MBE | Mean Bias Error |
| Md | Median |
| MOL-RAO | Meteorologisches Observatorium Lindenberg – Richard-Aßmann-Observatorium: DWD Observatory Lindenberg, Lindenberg (Tauche), Germany |
| NST | Normalized Spectral Transmittance |
| Ny-Å | Ny-Ålesund, Spitzbergen, Svalbard |
| OD | Optical Depth |
| ODF | Optical Depth Factor |





| | |
|---|---|
| PMOD/WRC | Physikalisch-Meteorologisches Observatorium Davos / World Radiation Center, Davos, Switzerland |
| QA | Quality Assurance |
| RGB | Red Green Blue |
| RIMO | ROLO Implementation for Moon photometry Observation: open lunar reflectance model developed by GOA and AEMET/IARC |
| ROD | Rayleigh Optical Depth |
| ROLO | Robotic Lunar Observatory: lunar reflectance model of United States Geological Survey (USGS) |
| SD | Standard Deviation |
| SMN | Servicio Meteorológico Nacional: National Meteorological Service of Argentina |
| SS | Star Signal |
| SZA | Solar Zenith Angle |
| TF | Tri-band Filter |
| TOC | Total Ozone Column (in DU) |
| TOD | Total Optical Depth |
| TWVC | Total Water Vapor Column |
| UGR | Universidad de Granada, Granada, Spain |
| USI | Uncalibrated Star Irradiance |
| $USI^0$ | Uncalibrated extraterrestrial Star Irradiance |
| UTC | Coordinated Universal Time |
| UVa | Universidad de Valladolid, Valladolid, Spain |




| Paramater | Description |
|---|---|
| $\alpha$ | Angström Exponent |
| $\beta$ | Angström turbidity |
| $\theta$ | Star zenith angle |
| $\tau$ | Total Optical Depth (TOD) |
| $\lambda$ | Wavelength |
| $\sigma$ | Standard Deviation (SD) |
| $\Delta$ | AOD differences |
| $\lambda_{eff}$ | Effective wavelength |
| $I(\lambda)$ | Spectral Irradiance (in $Wm^{-2}nm^{-1}$) |
| $m$ | Optical air mass |
| Md | Median |
| $r_w$ | Weighted correlation coefficient |
| $r$ | Correlation coefficient |
| $S(\lambda)$ | Spectral response |



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





**Table 1.** Detailed information about the location and measurement period of each GOA-SCAN all-sky camera used in this work. The acronyms in the table are defined as follows: SMN (Servicio Meteorológico Nacional; Argentinian Meteorological Service), AWI (Alfred Wegener Institute for Polar and Marine Research), GOA-UVa (Grupo de Óptica Atmosférica de la Universidad de Valladolid; Atmospheric Optics Group of the University of Valladolid), AEMet (Agencia Estatal de Meteorología; Spanish Meteorological Agency), ASC (Andøya Space Center), DWD (Deutscher Wetterdienst; German Meteorological Service), GFAT-UGR (Grupo de Física de la Atmósfera de la Universidad de Granada; Atmospheric Physics Group of the University of Granada), and PMOD-WRC (Physikalisch-Meteorologisches Observatorium Davos and World Radiation Center). Present corresponds to 22 January 2025.

| Camera ID | Camera Model | Start Date | End Date | Station | Country | Latitude (°N) | Longitude (°E) | Elevation (m asl) | Site Responsible | Camera Owner |
|---|---|---|---|---|---|---|---|---|---|---|
| C003 | OMEA-3C | 2018-01-26 | Present | Marambio station | Antarctica, Argentina | -64.240 | -56.625 | 200 | SMN | GOA-UVa |
| C004 | OMEA-3C | 2018-10-09 | Present | Ny-Ålesund | Norway | 78.923 | 11.923 | 38 | AWI | GOA-UVa |
| C005 | OMEA-3C | 2020-07-16 | 2021-09-27 | Valladolid | Spain | 41.663 | -4.706 | 710 | GOA-UVa | GOA-UVa |
| C005 | OMEA-3C | 2021-10-06 | 2022-01-24 | Fuencaliente | Spain | 28.487 | -17.849 | 630 | AEMet | GOA-UVa |
| C005 | OMEA-3C | 2022-02-04 | 2023-02-15 | Izaña | Spain | 28.309 | -16.499 | 2365 | AEMet | GOA-UVa |
| C005 | OMEA-3C | 2023-04-18 | Present | Andøya | Norway | 69.278 | 16.009 | 380 | ASC | GOA-UVa |
| C006 | OMEA-3C | 2020-07-29 | 2021-02-12 | Lindenberg | Germany | 52.209 | 14.121 | 120 | DWD | GOA-UVa |
| C009 | OMEA-3C | 2021-02-12 | Present | Lindenberg | Germany | 52.209 | 14.121 | 120 | DWD | GOA-UVa |
| C011 | OMEA-3C | 2021-10-05 | 2024-01-11 | Valladolid | Spain | 41.663 | -4.706 | 710 | GOA-UVa | GOA-UVa |
| C011 | OMEA-3C | 2024-03-04 | Present | Granada | Spain | 37.164 | -3.605 | 673 | GFAT-UGR | GOA-UVa |
| C013 | OMEA-3C-TF | 2023-06-27 | Present | Valladolid | Spain | 41.663 | -4.706 | 710 | GOA-UVa | GOA-UVa |
| C038 | OMEA-3C | 2024-04-19 | Present | Davos | Switzerland | 46.813 | 9.844 | 1584 | PMOD-WRC | PMOD-WRC |





**Table 2.** Median (Md) and standard deviation ($\sigma$) of the $log(USI^0)$ values for different stars and channels for all the measurement period of camera C011. The number of averaged data (N) is included.

| Star | Red N | Red Md | Red $\sigma$ | Green N | Green Md | Green $\sigma$ | Blue N | Blue Md | Blue $\sigma$ | Gray N | Gray Md | Gray $\sigma$ |
|---|---|---|---|---|---|---|---|---|---|---|---|---|
| Adhara | 27 | 8.21 | 0.66 | 50 | 9.06 | 0.18 | 55 | 8.78 | 0.30 | 60 | 8.90 | 0.21 |
| Aldebaran | 325 | 9.45 | 0.30 | 400 | 9.60 | 0.20 | 342 | 8.59 | 0.26 | 428 | 9.50 | 0.24 |
| Algol | 211 | 7.78 | 0.41 | 242 | 8.54 | 0.24 | 292 | 8.28 | 0.30 | 312 | 8.37 | 0.24 |
| Alhena | 188 | 7.92 | 0.25 | 273 | 8.71 | 0.17 | 309 | 8.40 | 0.22 | 324 | 8.54 | 0.16 |
| Alioth | 280 | 8.04 | 0.56 | 356 | 8.73 | 0.43 | 462 | 8.53 | 0.24 | 442 | 8.67 | 0.39 |
| Alkaid | 145 | 7.75 | 0.48 | 268 | 8.62 | 0.27 | 385 | 8.51 | 0.23 | 348 | 8.60 | 0.25 |
| Alnilam | 271 | 8.13 | 0.23 | 294 | 8.90 | 0.20 | 371 | 8.69 | 0.24 | 358 | 8.75 | 0.18 |
| Alnitak | 1 | 8.33 | 0.00 | 1 | 8.82 | 0.00 | 3 | 8.58 | 0.14 | 2 | 8.71 | 0.01 |
| Alphecca | 72 | 7.56 | 0.50 | 131 | 8.31 | 0.38 | 302 | 8.06 | 0.20 | 315 | 8.23 | 0.24 |
| Alpheratz | 201 | 7.78 | 0.69 | 264 | 8.62 | 0.36 | 309 | 8.32 | 0.22 | 330 | 8.42 | 0.33 |
| Altair | 219 | 9.10 | 0.32 | 361 | 9.67 | 0.26 | 399 | 9.38 | 0.31 | 389 | 9.55 | 0.24 |
| Antares | 197 | 9.43 | 0.26 | 182 | 9.47 | 0.26 | 148 | 8.25 | 0.30 | 226 | 9.42 | 0.23 |
| Arcturus | 111 | 10.27 | 0.34 | 174 | 10.31 | 0.33 | 111 | 9.65 | 0.44 | 286 | 10.36 | 0.27 |
| Bellatrix | 238 | 8.15 | 0.20 | 347 | 9.00 | 0.19 | 393 | 8.79 | 0.22 | 367 | 8.80 | 0.16 |
| Betelgeuse | 268 | 9.82 | 0.31 | 375 | 9.89 | 0.23 | 298 | 8.77 | 0.30 | 382 | 9.85 | 0.25 |
| Capella | 235 | 9.96 | 0.32 | 322 | 10.32 | 0.28 | 296 | 9.77 | 0.30 | 380 | 10.22 | 0.28 |
| Caph | 152 | 7.68 | 0.44 | 190 | 8.31 | 0.30 | 191 | 7.76 | 0.21 | 244 | 8.15 | 0.26 |
| Castor | 222 | 8.27 | 0.27 | 282 | 9.01 | 0.22 | 369 | 8.74 | 0.25 | 337 | 8.84 | 0.19 |
| Deneb | 98 | 8.34 | 0.27 | 157 | 9.18 | 0.21 | 302 | 8.90 | 0.23 | 208 | 9.04 | 0.29 |
| Dubhe | 237 | 8.42 | 0.68 | 264 | 8.71 | 0.54 | 265 | 7.95 | 0.25 | 340 | 8.66 | 0.51 |
| Elnath | 234 | 8.15 | 0.45 | 338 | 8.96 | 0.34 | 388 | 8.74 | 0.26 | 366 | 8.78 | 0.27 |
| Eta Herculis | 8 | 10.64 | 1.16 | 14 | 10.76 | 1.12 | 11 | 9.10 | 0.40 | 33 | 7.18 | 1.28 |
| Fomalhaut | 67 | 8.56 | 0.48 | 99 | 9.29 | 0.60 | 121 | 8.93 | 0.24 | 106 | 9.17 | 0.38 |
| Gamma Herculis | 27 | 7.34 | 0.63 | 7 | 7.80 | 0.70 | 6 | 7.01 | 0.23 | 119 | 7.68 | 0.25 |
| Markab | 94 | 7.41 | 0.51 | 189 | 8.21 | 0.45 | 213 | 7.87 | 0.19 | 270 | 8.04 | 0.26 |
| Merak | 124 | 7.66 | 0.96 | 185 | 8.20 | 0.62 | 285 | 7.98 | 0.24 | 302 | 8.12 | 0.52 |
| Mintaka | 195 | 7.78 | 0.31 | 245 | 8.81 | 0.31 | 340 | 8.27 | 0.28 | 271 | 8.65 | 0.42 |
| Mirfak | 212 | 8.24 | 0.26 | 208 | 8.74 | 0.27 | 222 | 8.19 | 0.22 | 271 | 8.69 | 0.21 |
| Mizar | 318 | 8.04 | 0.51 | 399 | 8.76 | 0.38 | 418 | 8.42 | 0.23 | 413 | 8.60 | 0.34 |
| Pollux | 271 | 9.01 | 0.28 | 322 | 9.33 | 0.28 | 350 | 8.68 | 0.25 | 385 | 9.23 | 0.23 |
| Procyon | 229 | 9.58 | 0.27 | 354 | 10.09 | 0.21 | 341 | 9.66 | 0.28 | 372 | 9.96 | 0.22 |
| Regulus | 190 | 8.42 | 0.36 | 276 | 9.21 | 0.24 | 335 | 8.97 | 0.27 | 319 | 9.05 | 0.26 |
| Rigel | 240 | 9.61 | 0.30 | 361 | 10.30 | 0.24 | 318 | 10.11 | 0.30 | 383 | 10.12 | 0.26 |
| Sheratan | 107 | 7.36 | 0.32 | 133 | 8.14 | 0.48 | 141 | 7.51 | 0.25 | 183 | 7.92 | 0.30 |
| Sirius | 36 | 11.12 | 0.58 | 53 | 11.90 | 0.37 | 17 | 11.64 | 0.84 | 85 | 11.64 | 0.48 |
| Spica | 268 | 8.75 | 0.21 | 319 | 9.59 | 0.20 | 352 | 9.50 | 0.27 | 352 | 9.40 | 0.26 |
| Vega | 164 | 9.53 | 0.40 | 307 | 10.28 | 0.31 | 329 | 10.18 | 0.34 | 367 | 10.18 | 0.25 |
| Wezen | 80 | 8.28 | 0.25 | 81 | 8.81 | 0.27 | 96 | 8.46 | 0.37 | 116 | 8.62 | 0.32 |



**Table 3.** Median of the calculated effective wavelengths (± standard deviation) for the four color channels of the OMEA-3C and OMEA-3C-TF camera models, calculated for different stars.

| Star | OMEA-3C | | | | OMEA-3C-TF | | | |
|---|---|---|---|---|---|---|---|---|
| | Red (nm) | Green (nm) | Blue (nm) | Gray (nm) | Red (nm) | Green (nm) | Blue (nm) | Gray (nm) |
| Achernar | 585.4±3.7 | 530.4±2.3 | 462.0±3.0 | 530.3±3.8 | 611.8±1.6 | 534.3±2.2 | 470.0±1.9 | 542.3±3.9 |
| Adhara | 584.8±3.8 | 530.2±2.3 | 461.5±3.0 | 529.9±3.9 | 611.6±1.7 | 534.2±2.2 | 469.9±1.9 | 542.1±3.9 |
| Alkaid | 585.3±3.7 | 530.6±2.3 | 462.0±3.0 | 530.5±3.8 | 611.5±1.7 | 534.0±2.2 | 469.8±1.9 | 541.7±3.9 |
| Alpheratz | 587.0±3.4 | 531.6±2.3 | 462.9±3.0 | 532.2±3.8 | 612.4±1.6 | 535.0±2.2 | 470.2±1.9 | 543.4±3.9 |
| Altair | 593.0±2.6 | 535.6±2.2 | 467.4±3.1 | 538.6±3.5 | 615.1±1.3 | 538.5±2.2 | 472.8±2.3 | 549.8±3.8 |
| Arcturus | 604.6±1.2 | 547.9±2.0 | 485.5±3.4 | 557.5±2.7 | 621.2±0.7 | 551.3±2.4 | 485.9±3.6 | 570.5±3.3 |
| Bellatrix | 584.8±3.8 | 530.2±2.3 | 461.7±3.0 | 530.0±3.9 | 611.4±1.7 | 534.0±2.1 | 469.7±1.9 | 541.7±3.9 |
| Canopus | 591.9±2.8 | 534.7±2.2 | 466.3±3.1 | 537.3±3.6 | 614.9±1.3 | 538.1±2.2 | 472.5±2.2 | 549.1±3.9 |
| Capella | 600.8±1.6 | 542.6±2.1 | 476.9±3.3 | 549.8±3.1 | 619.2±0.9 | 546.0±2.3 | 479.7±3.0 | 562.4±3.5 |
| Caph | 595.0±2.4 | 537.0±2.2 | 469.2±3.2 | 541.0±3.4 | 616.2±1.2 | 540.2±2.3 | 474.1±2.4 | 552.7±3.8 |
| Deneb | 591.4±2.9 | 534.4±2.2 | 466.0±3.1 | 536.9±3.6 | 614.6±1.3 | 537.9±2.2 | 472.6±2.2 | 548.7±3.8 |
| Dubhe | 603.3±1.3 | 545.7±2.0 | 482.5±3.4 | 554.5±2.8 | 620.4±0.7 | 549.1±2.3 | 483.7±3.4 | 567.2±3.4 |
| Elnath | 586.5±3.5 | 531.2±2.3 | 462.6±3.0 | 531.6±3.8 | 612.2±1.6 | 534.8±2.2 | 470.1±1.9 | 543.1±3.9 |
| Eta Herculis | 601.9±1.5 | 543.7±2.0 | 479.1±3.4 | 551.6±3.0 | 619.8±0.8 | 547.2±2.3 | 481.1±3.2 | 564.3±3.5 |
| Gacrux | 608.1±1.1 | 551.1±1.9 | 492.1±3.7 | 562.4±2.6 | 623.3±0.6 | 554.3±2.3 | 493.7±4.2 | 575.4±3.0 |
| Gamma Herculis | 594.1±2.5 | 536.7±2.2 | 468.4±3.2 | 540.3±3.5 | 615.8±1.2 | 539.5±2.2 | 473.7±2.4 | 551.5±3.8 |
| Kochab | 606.2±1.1 | 550.6±1.9 | 489.7±3.5 | 561.2±2.6 | 622.1±0.6 | 554.1±2.4 | 489.3±3.9 | 574.5±3.1 |
| Markab | 588.0±3.3 | 532.3±2.3 | 463.6±3.0 | 533.2±3.7 | 612.8±1.5 | 535.5±2.2 | 470.6±2.0 | 544.4±3.9 |
| Merak | 588.4±3.2 | 532.6±2.2 | 464.0±3.0 | 533.7±3.7 | 612.9±1.5 | 535.6±2.2 | 470.8±2.0 | 544.6±3.9 |
| Miaplacidus | 589.0±3.2 | 532.9±2.2 | 464.1±3.1 | 534.3±3.7 | 613.3±1.5 | 536.2±2.2 | 471.3±2.1 | 545.6±3.9 |
| Mirfak | 597.0±2.1 | 538.9±2.1 | 471.5±3.2 | 543.9±3.3 | 617.4±1.1 | 542.2±2.3 | 476.0±2.6 | 556.3±3.7 |
| Mizar | 589.1±3.1 | 533.1±2.2 | 464.5±3.0 | 534.5±3.7 | 613.2±1.5 | 535.9±2.2 | 471.0±2.0 | 545.2±3.9 |
| Pollux | 602.5±1.4 | 544.6±2.0 | 480.7±3.3 | 552.9±2.9 | 620.1±0.8 | 548.1±2.3 | 481.9±3.2 | 565.7±3.4 |
| Procyon | 596.1±2.3 | 538.0±2.2 | 470.3±3.2 | 542.6±3.4 | 616.9±1.1 | 541.3±2.3 | 475.2±2.5 | 554.7±3.7 |
| Regulus | 586.8±3.5 | 531.4±2.3 | 462.8±3.0 | 531.9±3.8 | 612.3±1.6 | 534.9±2.2 | 470.3±2.0 | 543.3±3.9 |
| Rigel | 589.0±3.2 | 532.8±2.3 | 464.0±3.1 | 534.2±3.7 | 613.6±1.4 | 536.5±2.2 | 471.6±2.1 | 546.3±3.8 |
| Sheratan | 591.2±2.8 | 534.4±2.2 | 466.1±3.1 | 536.6±3.6 | 614.1±1.4 | 537.1±2.2 | 471.8±2.1 | 547.3±3.9 |
| Sirius | 587.9±3.3 | 532.1±2.3 | 463.5±3.0 | 533.0±3.7 | 612.7±1.5 | 535.3±2.2 | 470.5±2.0 | 544.1±3.9 |
| Vega | 588.4±3.2 | 532.6±2.2 | 463.9±3.0 | 533.7±3.7 | 612.9±1.5 | 535.6±2.2 | 470.8±2.0 | 544.7±3.9 |
| Wezen | 598.7±1.9 | 540.5±2.1 | 474.5±3.3 | 546.6±3.2 | 618.3±1.0 | 543.9±2.3 | 477.9±2.8 | 559.1±3.6 |
| All | 591.3±7.1 | 534.4±6.3 | 466.1±8.9 | 536.7±9.9 | 614.4±3.6 | 537.5±6.3 | 472.2±6.4 | 548.0±10.5 |



**Table 4.** Mean values (± standard deviation) of the calculated Optical Depth Factors (ODF) of ozone (O3), nitrogen dioxide (NO2), water vapor (H2O) and Rayleigh scattering for the four color channels of the OMEA-3C and OMEA-3C-TF camera models.

| ODF | OMEA-3C | | | | OMEA-3C-TF | | | |
|---|---|---|---|---|---|---|---|---|
| | Red | Green | Blue | Gray | Red | Green | Blue | Gray |
| O3 (DU$^{-1}$) x10$^5$ | 10.0±0.3 | 7.0±0.5 | 2.0±0.5 | 7.1±0.7 | 9.1±0.1 | 6.5±0.3 | 1.9±0.4 | 6.7±0.5 |
| NO2 (DU$^{-1}$) x10$^4$ | 17.0±4.2 | 39.2±4.1 | 97.2±8.4 | 40.7±6.5 | 9.8±1.6 | 36.5±3.3 | 93.9±5.7 | 35.6±5.7 |
| H2O (mm$^{-1}$) x10$^5$ | 22.7±1.0 | 10.1±1.2 | 2.3±0.7 | 11.6±1.5 | 6.3±0.4 | 2.6±0.3 | 0.9±0.2 | 3.4±0.5 |
| Rayleigh (hPa$^{-1}$) x10$^5$ | 7.8±0.6 | 11.3±0.6 | 19.2±1.4 | 11.3±0.9 | 6.5±0.2 | 10.9±0.5 | 18.2±0.8 | 10.4±0.8 |





**Table 5.** Statistical estimators of the comparison between the AOD obtained from cameras and photometers for different cameras and channels. N represents the number of available data pairs, r is the correlation coefficient, and b and $y_0$ are the slope and y-intercept of the least-squares linear fit between the camera and photometer AODs. Regarding the differences between the AOD from the cameras and photometers ($\Delta$), the table includes the mean (Mean Bias Error, MBE), the standard deviation (SD), and the percentage of $\Delta$ absolute values that fall within one time the sum of the uncertainties of both AODs ($\Delta < \sigma$) and within two times this sum ($\Delta < 2\sigma$).

| Camera ID | Station | channel | N | r | MBE | SD | b | $y_0$ | $\Delta < \sigma$ (%) | $\Delta < 2\sigma$ (%) |
|---|---|---|---|---|---|---|---|---|---|---|
| C003 | Marambio | Red | 1276 | 0.350 | 0.035 | 0.074 | 0.385 | 0.072 | 63.1 | 87.0 |
| C003 | Marambio | Green | 1307 | 0.512 | 0.044 | 0.072 | 0.559 | 0.073 | 51.0 | 80.3 |
| C003 | Marambio | Blue | 903 | 0.184 | 0.049 | 0.091 | 0.229 | 0.097 | 57.9 | 78.1 |
| C003 | Marambio | Gray | 1298 | 0.554 | 0.040 | 0.071 | 0.645 | 0.063 | 54.6 | 83.0 |
| C004 | Ny-Ålesund | Red | 6997 | 0.563 | 0.024 | 0.037 | 0.703 | 0.037 | 78.4 | 95.6 |
| C004 | Ny-Ålesund | Green | 8985 | 0.521 | 0.020 | 0.041 | 0.669 | 0.036 | 68.2 | 93.7 |
| C004 | Ny-Ålesund | Blue | 5555 | 0.146 | 0.014 | 0.060 | 0.230 | 0.058 | 49.6 | 87.4 |
| C004 | Ny-Ålesund | Gray | 9197 | 0.504 | 0.016 | 0.041 | 0.627 | 0.034 | 66.2 | 94.5 |
| C005 | All | Red | 8334 | 0.915 | 0.016 | 0.024 | 0.963 | 0.018 | 83.0 | 98.5 |
| C005 | All | Green | 11840 | 0.977 | 0.030 | 0.020 | 1.019 | 0.029 | 45.7 | 96.2 |
| C005 | All | Blue | 11749 | 0.962 | 0.024 | 0.026 | 0.951 | 0.028 | 57.8 | 95.8 |
| C005 | All | Gray | 12273 | 0.974 | 0.027 | 0.021 | 0.985 | 0.028 | 53.4 | 96.3 |
| C006 | Lindenberg | Red | 295 | 0.890 | 0.008 | 0.034 | 0.861 | 0.022 | 84.7 | 98.3 |
| C006 | Lindenberg | Green | 683 | 0.955 | 0.028 | 0.028 | 0.875 | 0.044 | 70.7 | 98.2 |
| C006 | Lindenberg | Blue | 502 | 0.959 | 0.016 | 0.032 | 0.886 | 0.035 | 83.1 | 99.0 |
| C006 | Lindenberg | Gray | 651 | 0.948 | 0.026 | 0.026 | 0.925 | 0.035 | 72.8 | 98.2 |
| C009 | Lindenberg | Red | 6461 | 0.912 | 0.007 | 0.029 | 0.902 | 0.016 | 88.9 | 98.4 |
| C009 | Lindenberg | Green | 9693 | 0.940 | 0.014 | 0.030 | 0.927 | 0.023 | 83.3 | 97.2 |
| C009 | Lindenberg | Blue | 8911 | 0.932 | -0.004 | 0.033 | 0.926 | 0.006 | 88.0 | 97.7 |
| C009 | Lindenberg | Gray | 10238 | 0.951 | 0.012 | 0.028 | 0.917 | 0.022 | 84.4 | 97.0 |
| C011 | All | Red | 7320 | 0.919 | 0.006 | 0.033 | 0.907 | 0.013 | 86.9 | 99.0 |
| C011 | All | Green | 10318 | 0.923 | 0.011 | 0.036 | 0.824 | 0.026 | 74.1 | 97.5 |
| C011 | All | Blue | 11679 | 0.768 | 0.015 | 0.054 | 0.640 | 0.050 | 52.2 | 91.4 |
| C011 | All | Gray | 18080 | 0.948 | 0.022 | 0.029 | 0.849 | 0.035 | 60.5 | 93.9 |
| C013 | Valladolid | Red | 7162 | 0.897 | 0.036 | 0.024 | 0.980 | 0.037 | 55.9 | 97.1 |
| C013 | Valladolid | Green | 9356 | 0.949 | 0.026 | 0.018 | 0.932 | 0.031 | 69.4 | 99.0 |
| C013 | Valladolid | Blue | 6533 | 0.896 | 0.043 | 0.023 | 0.930 | 0.048 | 40.8 | 95.1 |
| C013 | Valladolid | Gray | 9621 | 0.962 | 0.040 | 0.016 | 0.951 | 0.044 | 25.3 | 93.6 |
| C038 | Davos | Red | 579 | 0.706 | 0.049 | 0.037 | 0.889 | 0.055 | 43.5 | 91.9 |
| C038 | Davos | Green | 868 | 0.864 | 0.042 | 0.030 | 0.936 | 0.046 | 47.2 | 95.2 |
| C038 | Davos | Blue | 752 | 0.783 | 0.044 | 0.039 | 0.893 | 0.051 | 50.7 | 91.6 |
| C038 | Davos | Gray | 1099 | 0.882 | 0.043 | 0.024 | 1.030 | 0.042 | 41.5 | 96.7 |
| All | All | Red | 38424 | 0.856 | 0.019 | 0.035 | 0.875 | 0.027 | 77.6 | 97.3 |
| All | All | Green | 53050 | 0.917 | 0.022 | 0.033 | 0.904 | 0.029 | 66.6 | 96.3 |
| All | All | Blue | 46584 | 0.847 | 0.019 | 0.045 | 0.806 | 0.037 | 59.0 | 93.6 |
| All | All | Gray | 62457 | 0.928 | 0.024 | 0.031 | 0.896 | 0.033 | 58.1 | 94.8 |



**Table 6.** Same values as in Table 5, but for the different locations of cameras C005 and C011 separately.

| Camera ID | Station | channel | N | r | MBE | SD | b | $y_0$ | $\Delta < \sigma$ (%) | $\Delta < 2\sigma$ (%) |
|-----------|---------|---------|-----|-----|-----|-----|-----|-----|-----|-----|
| C005 | Izaña | Red | 5044 | 0.928 | 0.018 | 0.019 | 1.037 | 0.017 | 81.6 | 99.1 |
| C005 | Izaña | Green | 6661 | 0.988 | 0.030 | 0.011 | 1.042 | 0.028 | 37.3 | 98.3 |
| C005 | Izaña | Blue | 6637 | 0.978 | 0.032 | 0.015 | 1.025 | 0.030 | 40.3 | 95.5 |
| C005 | Izaña | Gray | 6719 | 0.989 | 0.027 | 0.010 | 1.022 | 0.026 | 45.8 | 98.3 |
| C005 | Valladolid | Red | 1740 | 0.917 | 0.011 | 0.029 | 0.982 | 0.013 | 85.9 | 97.5 |
| C005 | Valladolid | Green | 3266 | 0.976 | 0.032 | 0.025 | 1.011 | 0.030 | 54.7 | 93.9 |
| C005 | Valladolid | Blue | 3146 | 0.965 | 0.012 | 0.031 | 0.973 | 0.016 | 84.0 | 96.5 |
| C005 | Valladolid | Gray | 3477 | 0.973 | 0.027 | 0.025 | 0.978 | 0.030 | 63.0 | 94.4 |
| C005 | Fuencaliente | Red | 544 | 0.499 | 0.012 | 0.030 | 0.543 | 0.037 | 85.5 | 96.3 |
| C005 | Fuencaliente | Green | 743 | 0.971 | 0.038 | 0.028 | 1.001 | 0.038 | 42.4 | 90.6 |
| C005 | Fuencaliente | Blue | 830 | 0.949 | 0.029 | 0.032 | 0.915 | 0.038 | 56.6 | 93.1 |
| C005 | Fuencaliente | Gray | 893 | 0.959 | 0.034 | 0.032 | 0.931 | 0.040 | 41.8 | 88.7 |
| C005 | Andøya | Red | 1006 | 0.895 | 0.016 | 0.031 | 0.890 | 0.022 | 83.5 | 98.3 |
| C005 | Andøya | Green | 1170 | 0.880 | 0.025 | 0.033 | 0.944 | 0.028 | 70.1 | 94.4 |
| C005 | Andøya | Blue | 1136 | 0.863 | 0.006 | 0.036 | 0.870 | 0.013 | 87.9 | 97.4 |
| C005 | Andøya | Gray | 1184 | 0.903 | 0.020 | 0.035 | 0.905 | 0.025 | 76.9 | 95.8 |
| C011 | Granada | Red | 1914 | 0.959 | 0.002 | 0.036 | 0.951 | 0.008 | 91.3 | 99.1 |
| C011 | Granada | Green | 2645 | 0.965 | 0.005 | 0.032 | 0.914 | 0.015 | 83.1 | 98.1 |
| C011 | Granada | Blue | 2539 | 0.835 | -0.013 | 0.047 | 0.821 | 0.007 | 68.8 | 95.1 |
| C011 | Granada | Gray | 3999 | 0.970 | 0.013 | 0.025 | 0.958 | 0.018 | 82.3 | 98.0 |
| C011 | Valladolid | Red | 5406 | 0.824 | 0.007 | 0.032 | 0.823 | 0.018 | 85.4 | 99.0 |
| C011 | Valladolid | Green | 7673 | 0.876 | 0.014 | 0.036 | 0.745 | 0.032 | 71.0 | 97.3 |
| C011 | Valladolid | Blue | 9140 | 0.772 | 0.023 | 0.053 | 0.611 | 0.059 | 47.6 | 90.3 |
| C011 | Valladolid | Gray | 14081 | 0.942 | 0.024 | 0.030 | 0.811 | 0.040 | 54.4 | 92.8 |