# Peer review of "Star photometry with all-sky cameras to retrieve aerosol optical depth at night-time"

_EGUsphere, 2025_

## Author Response (AR1)

**Response to the Referee (Liviu Ivanescu) comments for the manuscript "Star photometry with all-sky cameras to retrieve aerosol optical depth at night-time" By Roberto Román et al. in AMT**

First of all, we would like to thank the time and effort of the referee for the detailed review of the manuscript. Reviewer comments (RC) are in black font and author comments (AC) are in red.

**Author's answer to Referee Liviu Ivanescu**

**General comments:**
RC: This study addresses a technical approach that potentially renders star-photometry cheaper to manufacture and to operate unattended. It implies a comprehensive and complex analysis, and the authors should be commended for completing it. The results are meaningful and encouraging. The paper is well written, and the figures are of good quality. It represents an important development, and I recommend it for publication!
AC: We sincerely appreciate this comment. We think it perfectly captures the essence of the paper.

**Specific comments:**
RC: Being the first such development properly analysed, and given the scarcity of operational star-photometers, the retried optical depth had to be compared with that of moon-photometers. Their accuracy remains however questionable, partly due to the ROLO model accuracy, and being prone to forward scattering in the presence of cirrus clouds or PSCs, that are very difficult to screen out with standard algorithms. I would advise therefore pursuing the analysis by also comparing the results with the available star-photometers in Lindenberg and Ny-Alesund. They can identify such clouds with a spectral screening (O'Neill 2016, doi: 10.5194/acp-16-12753-2016).
AC: As rightly pointed out by the reviewer, the scarcity of operational star-photometers has limited our comparison to measurements obtained with moon-photometers. We agree that the accuracy of the ROLO model (or its implementation in the RIMO algorithm) appears to affect the accuracy of AOD retrievals from moon-photometers. However, recent studies (Román et al., 2020; González et al., 2020; AERONET Lunar AOD V3 Data 6 August 2024 webpage note) suggest that empirical corrections to ROLO/RIMO have led to significant improvements in the AOD accuracy with moon-photometers.

We also agree that high-cloud screening is more difficult at night with moon-photometers. For example, lunar aureole measurements are not available as solar aureole measurements are during the day. This limitation may lead to cloud-contaminated data in the comparison, as can be seen in Figure 12 under the presence of PSCs. We therefore agree that it is important to compare with star-photometers, not only because of their ability to perform spectral cloud screening, but also to evaluate how well the proposed method performs during nights without moon presence. To emphasize this point, the following sentence has been added to the conclusions:
"*In this regard, the proposed methodology should also be compared with measurements from star-photometers, such as those installed in Lindenberg or Ny-Ålesund, in order to assess the performance of the AOD retrieved with the all-sky cameras during moonless periods. This comparison could also benefit from the cloud-screening algorithms for ice clouds that have been developed for these instruments (Baibakov et al., 2015; O'Neill et al., 2016).*"

RC: Also for a future analysis, instead of using effective wavelength and effective optical depth, one may take advantage of a modified The Beer-Lambert-Bouguer law, specifically developed for wide filters, as in Rufener 1986 (http://adsabs. harvard.edu/abs/1986A&A...165..275R), as it's the case for most of the all-sky cameras used here.

AC: We decided to use the Beer-Lambert-Bouguer law and the Langley technique instead of other approaches, such as the one proposed by Rufener (1986), based on our prior experience with this methodology, particularly in solar and lunar photometry. Nevertheless, the idea of testing alternative methods is certainly interesting, and we think that the modified Beer-Lambert-Bouguer law mentioned by the reviewer should be explored in future work. We consider that our study opens the door to the use of all-sky cameras for star photometry. Even though our proposed method shows promising results, we agree that different methodologies should be investigated.

To clarify that, the beginning of Section 3.4 has been improved as:

"*The direct solar, lunar, or stellar irradiance at one single wavelength reaching the Earth is related to the atmospheric total optical depth (TOD) through Beer-Lambert-Bouguer law. This law is monochromatic, and there are modifications based on approximations for irradiance measurements performed with broadband filters (Rufener, 1986); however, in this work, the measured irradiance has been approximated as the irradiance at an effective wavelength (see Section 3.5) that satisfies the Beer-Lambert-Bouguer law. For stars, this relationship can be expressed in terms of the natural logarithm, as shown in Equation 3:*"

RC: The several subsequent screen algorithms to remove outliers may remove legitimate data points and underestimate the measurement uncertainty. For example, using several sigma as a criteria to remove outliers may not be appropriate in a non-gaussian, or non-symmetrical distribution. In this sense, particularly problematic may be the 1% constraint in the Langley fit. You may want to comment on this.

AC: We agree with the reviewer that several data-screening steps are involved in the processing. We expect that a simpler and more robust filtering approach could be developed in the future. Sigma criteria are technically valid only for gaussian, but we decided to make this approach. It is true that some of the discarded data points may in fact be valid; however, in the absence of more advanced or reliable screening algorithms, we have chosen to adopt a conservative approach, favouring the removal of potentially valid data rather than retaining data that could be affected by undetected issues.

Regarding the 1% criterion in the Langley fits, this threshold was established manually after visually inspecting a large number of Langley plots. In practice, it is not an especially strict condition, since a 1% relative error in ln(USI) values, which typically range between 7 and 10, corresponds to values of about 0.07 to 0.1. These tolerances are reasonably permissive for the purposes of this study.

The sentence of the reviewer has been added at the end of the Section 4 in the new manuscript:

"*The several subsequent screen algorithms to remove outliers may remove legitimate data points and underestimate the measurement uncertainty*".

RC: From my calculations, even with longer integration times, this camera should feature about 10 times more scintillation noise than the Lindenberg star-photometer. Therefore, the variability of the individual measurements in your Fig 8 may be due to this effect. Your filter may then simply select stars near zenith, that are less affected by scintillation. You may want to check and comment on this.

AC: Although scintillation may indeed affect the measurements, we think that what is shown in Figure 8 is not exactly as described by the reviewer. The variability of individual data points (visible as large oscillations) actually occurs at low zenith angles, that is, close to the zenith. This is the opposite of what the reviewer suggests, as the noise increases precisely when scintillation is expected to be lower. Therefore, the cause of this variability must be related to other effects, since it does not appear at higher zenith angles.

What we observe is that, at low zenith angles (i.e., low optical air masses), the pixel field of view is larger, which results in more photons reaching these pixels, increasing the likelihood of saturation. In addition, the number of pixels capturing the star image tends to be lower. Both effects reduce the number of valid measurements available at low zenith angles, leading to a low $N_{smooth}$ value and, consequently, a noisier average.

All these aspects are discussed in the manuscript, as can be seen in the following text:

"*Moreover, the AOD for some stars exhibits pronounced oscillations, such as Mirfak in the middle of the night or Pollux at the end. These oscillations primarily occur under conditions of low $N_{smooth}$ values and low optical air mass. This suggests that the oscillations may be caused by the insufficient number of USI values in the smoothing process described in Section 3.3. The lack of data could be partially related to low optical air masses, where the field of view of these pixels is larger due to the fisheye lens projection. This causes starlight to be concentrated in fewer pixels at low zenith angles, leading to the star segment not occupying enough pixels to be considered. Additionally, the higher concentration of starlight makes the signal in these pixels more prone to saturation under these low optical air mass conditions.*"

**Technical questions:**
RC: 16 - If the precision uncertainty is 0.03-0.04, the 0.02 accuracy uncertainty may not be that relevant. In addition, I think the moon photometer accuracy may have larger uncertainties than 0.02 anyway, as it's not a perfect reference to compare to anyway.

AC: Although the precision is larger than the accuracy, we do not consider the accuracy value to be negligible, as it provides useful information about whether the method tends to under- or overestimate the reference values. A precision of 0.03 with an accuracy of 0.00 gives a confidence interval of (−0.03, 0.03), while a precision of 0.03 with an accuracy of 0.02 results in a confidence interval of (−0.01, 0.05). For this reason, we believe it is appropriate and informative to report both metrics.

Regarding the moon photometer, we agree that it may not be a perfect reference, but it is often the only available source of night-time AOD measurements at most stations. In this context, it remains the most suitable reference for comparison. Román et al. (2020) found a precision of approximately 0.03 for moon photometer AOD when using a star photometer as reference. However, we believe that the actual uncertainty of the moon photometer AOD values may be lower, since the star photometer data used in that study presented some technical issues that could have led to larger differences than the true ones.

Anyway, we have changed the uncertainty on moon-photometer to 0.03 in the new manuscript version:

"*In this case, the uncertainty associated with the photometer AOD is assumed to be 0.03 based on the results of Román et al. (2020) but it could actually be lower*".

RC: 36 – "This methodology is followed by AERONET" – I would rather say "used" instead of "followed", as the latter suggests that AERONET follows a method invented by (Toledano et al., 2018), which it's not.
AC: "followed" has been replaced by "used" as this comment suggests.

RC: 136 – "without demosaicing neither white-balance correction" - I don't understand what this means.
AC: What we meant by that statement is that the pure signal of each pixel is stored, without applying any kind of demosaicing or white-balance. Each pixel measures the signal in only one color channel, and demosaicing refers to the interpolation technique used to estimate the values of the other two color channels for each pixel. White balance, on the other hand, consists of multiplying the values of each color channel by a specific factor to produce a more (visually) realistic image, a purely aesthetic adjustment.

Some all-sky camera models allow storing the raw pixel values after they are already affected by a prior white-balance correction. This was the case, for example, in Román et al. (2022), where the signal in the blue pixels was multiplied by a factor greater than two before being saved in raw format. Unfortunately, this caused many pixels that were not originally saturated to become saturated due to the applied white-balance.

To avoid this problem, we have configured all our cameras to store the rawest possible signal, without any such previous corrections.

RC: 165 – "long time exposures, star scintillation caused by atmospheric turbulence" cannot spread the light on several pixels. "Star scintillation" only varies the amplitude of the star irradiance, while the "atmospheric turbulence" can only spread the light to ~<20 arcsec, i.e. way less than the camera pixels (5.4 arcmin ~= 324 arcsec). Therefore, a "long time exposure" should not spread the light over several pixels, unless other effects may contribute, like instrument vibrations etc. The main reason should be "the camera's point spread function".
AC: We agree with this comment, and we have removed the reference to star scintillation from the sentence. However, long exposure times cannot be neglected because, from the camera's coordinate system, the camera itself does not move during the entire integration time, while the stars do. The camera is not like a star photometer that tracks the star's position. For example, in Valladolid on November 23, 2024, at 19:14:02 UTC, the zenith and azimuth angles of Altair are 54.069439º and 247.159338º, respectively. Just 34 seconds later, at 19:14:36 UTC, these angles are 54.167286º and 247.283492º, respectively. This corresponds to a difference of approximately 8.4 arcmin in about 30 seconds. In fact, the position of the C013 camera pixel corresponding to Altair's position at the start is (400, 909), while its position at the end is (399, 910).

The sentence has been modified in the new manuscript as follows:

"*Stars are punctual sources of light; however, their image in all-sky photos spans several pixels due to long time exposures (as stars move during the integration time), and primarily due to the camera's point spread function (Piotrowski et al., 2013).*".

RC: 323 – Using "log" symbol for natural logarithm may be used only when the use of natural logarithm is implicit. Since for star magnitudes is usually used log in base 10, the natural base is therefore not implicit. I would recommend using "ln" for natural logarithm to avoid confusion.

AC: It is true. In order to avoid confusion, we have replaced "log" by "ln" in the new manuscript version.

RC: In addition, the "monochromatic" equation (3) may be used borderline to the camera OMEA-3C-TF, as the filters have usually smaller bandwidth than 40 nm (as specified in section 6.1 of Ivanescu et al, 2021, https://doi.org/10.5194/amt-14-6561- 2021). However, for the OMEA-3C, one certainly needs to consider a wide band-equation, like the equation (2) of Rufener 1986 (http://adsabs. harvard.edu/abs/1986A&A...165..275R). This was especially developed for such filters. Your "effective wavelength" solution seems an oversimplification. While it is not necessary to change your study to accommodate this formula, I think it's necessary however to add a comment concerning this aspect.

AC: We totally agree, and it has been discussed above. The paragraph was modified as next:

"*The direct solar, lunar, or stellar irradiance at one single wavelength reaching the Earth is related to the atmospheric total optical depth (TOD) through Beer-Lambert-Bouguer law. This law is monochromatic, and there are modifications based on approximations for irradiance measurements performed with broadband filters (Rufener, 1986); however, in this work, the measured irradiance has been approximated as the irradiance at an effective wavelength (see Section 3.5) that satisfies the Beer-Lambert-Bouguer law. For stars, this relationship can be expressed in terms of the natural logarithm, as shown in Equation 3:*"

RC: 357 – "Table 2" – sigma/sqrt(N) gives uncertainties that propagates into OD of about 0.02 to 0.04. Since this is for the entire ~3 year period, for one year should lead (multiplied by sqrt(3))) to 0.03-0.07 uncertainty only due to log(USI0). Why doesn't one see this in the final claimed 0.02 accuracy or 0.02-0.04 precision?

AC: This is because the final claimed accuracy of 0.02 and precision of 0.02-0.04 are calculated with respect to the AOD reference values measured by the moon photometer. However, the propagated uncertainty on the AOD calculated with the proposed methodology is different (see the error bars in Figure 9).

We have followed the method suggested by the reviewer to calculate the sigma/sqrt(N) values for one year (with N divided by 3) using the data shown in Table 2. The following Figure R1 presents a histogram of the obtained values for all the stars. The most frequent values are 0.02 and 0.03, except for the Red channel (which has higher uncertainty, as seen, for example, in Figure 9). Additionally, this uncertainty must be divided by the optical airmass when propagated to the OD values. Finally, this uncertainty is averaged using all the different stars, which explains the behaviour of the propagated uncertainty shown in Figures 9, 11, and 12.

The $\Delta < \sigma$ and $\Delta < 2\sigma$ values were calculated to verify whether this propagated uncertainty aligns with the expected values when compared with other AOD sources, such as the moon photometer.

[Figure]

*Figure R1: Frequency histogram of the sigma/sqrt(N/3) values from Table 2 for each color channel. The x-axis is truncated at 0.1, as only a few values exceed this limit and are not shown.*

RC: 397 – "longer values"?
AC: It has been replaced by "longer wavelengths".

RC: 446 – The climate values for ozone and especially water vapor may be much further away from the actual values, than the value of the aerosol optical depth, leading to high uncertainties. How to you account for such uncertainties? On the other hand, TWVC is not necessary for OMEA-3C-TF, as all the water absorption bands are outside of its filters.
AC: It is true that the reanalysis data used for ozone and water vapor have uncertainties that may significantly differ from the actual values. However, we decided to use these values as they are more operational and widely available at all stations, most of which do not have real measurements of these gases. Unfortunately, we are not accounting for these uncertainties in the propagation of AOD uncertainty.

Regarding water vapor, the contribution of its uncertainty is likely to be insignificant. For example, assuming an uncertainty of about 2.5 mm (one of the highest uncertainties found by Antuña-Marrero et al., 2025*), it corresponds to a variation of only about 0.0005 in the Red channel optical depth, which is negligible. This uncertainty is even smaller for the OMEA-3C-TF model, as the reviewer points out, due to its lower sensitivity to water vapor absorption, as shown in Figure R2. Nevertheless, even though the sensitivity to water vapor absorption in OMEA-3C-TF is very low, we still decided to calculate and remove it from the TOD in the same manner as with the OMEA-3C model.

In the case of ozone, a variation of 100 DU corresponds to a variation in OD of about 0.01 in the worst-case scenario (Red channel). If we assume an uncertainty of around 15 DU, as the RMSE found by Wang et al. (2021**) for a couple of polar stations, this represents an uncertainty of about 0.0015 in the optical depth, which is also negligible when compared to the overall uncertainty.

We have added in the new manuscript the next sentence at the end of Section 3.6:

*"The uncertainty in these downloaded values is not considered and is therefore not propagated in the final uncertainty of the mentioned optical depths."*

[Figure]

*Figure R2: Optical depth of ozone ($\tau_{O3}$), nitrogen dioxide ($\tau_{NO2}$), water vapor ($\tau_{H2O}$), and Rayleigh scattering ($\tau_R$) as a function of total ozone column (TOC; panel a), NO2 column (panel b), total water vapor column (TWVC; panel c), and ground atmospheric pressure (panel d), respectively. These values are calculated for OMEA-3C-TF camera model, for the star Regulus with a zenith angle of 50°, and an Ångström exponent and turbidity of 1 and 0.11, respectively. The data are shown separately for the four color channels, with solid lines indicating the least-squares linear fits.*

*\*Antuña-Marrero et al.: Comparing integrated water vapor sun photometer observations over the Arctic with ERA5 and MERRA-2 reanalyses. Journal of Geophysical Research: Atmospheres, 130, e2024JD041120. https://doi.org/10.1029/2024JD041120. 2025.*
*\*\*Wang et al.: Evaluating the performance of ozone products derived from CrIS/NOAA20, AIRS/aqua and ERA5 reanalysis in the polar regions in 2020 using ground-based observations. Remote Sensing, 13(21), 4375. 2021.*

RC: 466 – "attributed to inaccuracies in the log(USI0)" – not sure why mentioning only this one in particular.

AC: Here we are referring to two different issues: (1) the noisy pattern we observed at low zenith angles, as discussed above, and (2) values that systematically deviate from those of the other stars. In this part, we focus on the second issue. For instance, in the case of Mirfak at 591 nm: around midnight, this star reached low zenith angles, which led to noisy measurements. However, even with this noise, the average AOD values should still be centered around 0.05, similar to those of other stars. Instead, the average values exhibit a bell-shaped curve, which suggests the presence of a fictitious cycle. These kinds of fictitious cycles are typically caused by calibration errors (Cachorro et al., 2004\*); in our case, likely due to an incorrect value of log(USI0); ln(USI0) in the new manuscript version.

*Cachorro et al.: The fictitious diurnal cycle of aerosol optical depth: A new approach for "in situ" calibration and correction of AOD data series. Geophysical research letters, 31(12). 2004.*

RC: 565 – "calibration values log(USI0) [...] may not be optimal for all stations". This should not depend on site location or local environmental conditions. USI0 is characteristic to a star and a camera. Some stars are variable stars. Also, the camera contribution may very in time due to optical and electronic throughput changes.
AC: We totally agree with the reviewer. The actual value of ln(USI0) (log(USI0) in the previous version) should not depend on the location. Changes in ln(USI0) can be due to variable stars or to camera degradation and/or variations over time. In fact, we proposed the dynamic use of ln(USI0) as a temporal moving average to account for potential changes in the stars or in the camera properties.

What we intended to explain is that, although ln(USI0) itself should not vary with location, the accuracy of its estimation using the Langley method may vary depending on the site. For example, a location with a more stable atmosphere will allow a more accurate Langley fit than one with significant aerosol variability. However, we acknowledge that the sentence in the previous manuscript was not sufficiently clear. Therefore, we have removed it in the new revised version.

RC: 576 – "precisionranges" must be "precision ranges"
AC: This typo has already been corrected in the new manuscript.

RC: 577 – "The lowest SD values are found for camera C013, which is the only one with a triband filter", again, the best result obtained with this camera (including smaller $y_0$ and b$\sim$=1) may be linked to its quasi-monochromatic filters.
AC: Yes, we agree. That was exactly what we intended to highlight: narrower filters lead to better results because they are closer to a monochromatic response. To clarify this point, we have rewritten the sentence following the reviewer's suggestion:

"*The lowest SD values are found for camera C013, which may be linked to its quasi-monochromatic filters.*"

RC: 584 – "The standard uncertainty is defined as the sum of the errors of the camera and photometer" – this may be true only for bias. For random errors one should add them quadratically.
AC: It is true. We have changed this in the new manuscript:

"*The standard uncertainty is defined as the square root of the sum of the squares of the errors of the camera and photometer AODs, while the expanded uncertainty is twice the standard uncertainty.*"

The $\Delta < \sigma$ and $\Delta < 2\sigma$ values have been recalculated taking this into account and the uncertainty on the moon photometer as 0.03. The final results are quite similar to the previous ones.

RC: 601 – "Andoya is a less suitable site for Langley calibration due to its higher latitude" – this is true only for some stars. Andoya should not be that far North (like Ny-Alesund is) in order to have air mass coverage issue for most the stars. Even for Ny-Alesund, probably half of the stars still cover the 2-5 airmass range in one night. Beyond this, the filtering algorithm discards those stars that don't cover the required range. Then, why the higher latitude may not be good for Langley calibration? The unstable atmosphere? Why should a higher latitude have an unstable atmosphere? Do you have a reference on that?

AC: Maybe this point is not clearly stated in the manuscript. We do not mean that atmospheres are more unstable at high latitudes. What we mean is that the change in optical air mass is slower at higher latitudes. Therefore, although this depends on each specific star, reaching a variation of 2 in optical air mass generally takes longer at higher latitudes. This means that the time required to perform a Langley plot is also longer. As a consequence, the probability of experiencing some atmospheric variation during this longer period increases, which may reduce the accuracy of the calibration.

A new sentence has been added in a previous paragraph to clarify this:

"*In the case of the camera C004 in Ny-Ålesund, the differences between the camera and the photometer may partly be due to the Langley calibration method, which is ideal for low-latitude regions but performs worse at high latitudes due to slower changes in the optical air mass. This means that the time required to perform a Langley plot is longer at high latitudes, which increases the probability of experiencing some atmospheric variation during this extended period, potentially reducing the accuracy of the calibration.*"

RC: 604 – "estimating AOD appears to depend on the location where they are installed" – Why? Couldn't be because of differences in available data?

AC: Although the true value of ln(USI0) should not depend on the location, as previously discussed, the accuracy of the ln(USI0) can indeed be influenced by the site where the Langley calibrations are performed. For instance, Izaña is one of the best locations in the world for carrying out Langley plots, which results in highly accurate values of ln(USI0) at that site. This can be observed in the low standard deviation reported for Izaña in Table 6. In fact, this result highlights the potential of periodically calibrating all-sky cameras at reference sites such as Izaña (similar to the AERONET reference photometers) as a powerful alternative to obtain accurate ln(USI0) values, rather than relying solely on the on-site calibration proposed in this paper. It was briefly mentioned in the conclusions.

Another factor that may vary with location is the surrounding environment, particularly what is visible around the camera's horizon. For example, in some stations there are nearby light sources that cause certain areas of the night-time images to be permanently saturated, rendering those portions unusable. In other stations, such conditions do not occur.

Therefore, while the intrinsic value of ln(USI0) is location-independent, there are indeed several site-dependent factors that can affect the accuracy of the AOD retrieval. Anyway, part of the observed differences could be because of differences in available data. The next sentences have been added at the end of Section 4.1:

"*Although part of the observed differences may be due to differences in the available data, the differences between locations are most likely related to how suitable each site is for obtaining highly accurate values of ln(USI0) through the Langley technique. This depends on whether the site exhibits a stable atmosphere and is geographically located in a region where the star optical air masses change rapidly enough for the atmospheric conditions to remain nearly constant during the Langley calibration (typically at low and mid-latitudes).*"

RC: 630 – "the camera AOD at 466 nm shows lower values compared to the photometer"- this may be explained by the forward scattering error due to the larger camera FOV (see section 6.3 of Ivanescu et al, 2021, https://doi.org/10.5194/amt-14-6561- 2021). The forward scattering brings more light into the camera and the OD appears smaller. This should be more evident in the blue/UV, where the aerosol scattering is higher.

AC: It is true that forward scattering contributes to an increase in the light within the star aureole, which is well characterized in the paper cited by the reviewer. However, we do not think that the differences observed in the blue channel in Figure 12 are due to this effect, or at least not primarily. According to lidar measurements, no coarse particles such as PSCs were present on the 22nd to introduce sufficient forward scattering in the measured signal that could explain differences greater than 100% between the photometer and the camera. Moreover, such a phenomenon should also affect the moon photometer measurements, which would then be expected to measure more light than predicted.

On the other hand, if we consider the forward scattering hypothesis, even though it is more pronounced at 466 nm, its effect should also be visible at other wavelengths such as 534 nm. However, this is not the case, since at that wavelength the camera slightly overestimates the photometer AOD.

For all these reasons, we find the hypothesis of an insufficiently accurate calibration through the Langley technique more plausible.

**Response to the Anonymous Referee #2 comments for the manuscript "Star photometry with all-sky cameras to retrieve aerosol optical depth at night-time" By Roberto Román et al. in AMT**

First of all, we would like to thank the time and effort of the referee for the detailed review of the manuscript. Reviewer comments (RC) are in black font and author comments (AC) are in red.

**Author's answer to Anonymous Referee #2**

***General comment***

RC: Strengths:
  - **Innovation**: The study proposes all-sky cameras for star photometry, addressing the nighttime AOD data gap.
  - **Cost-Effectiveness**: Using commercial imaging devices could make aerosol monitoring more accessible if proven robust.
AC: We think the reviewer remarked two really important points of our work. We would also like to thank the reviewer for highlighting the strengths of the work in each section.

RC: Analysis:
  - **Assumptions**: Star intensity stability and uniform aerosol distribution are supposed, but intrinsic variability and atmospheric inhomogeneity could introduce biases.
  - **Comparative Advantage**: The study lacks discussion on trade-offs in spectral response, sensitivity, and precision compared to dedicated photometers.
AC: We acknowledge the limitations associated with the assumptions of star irradiance stability and spatial homogeneity of aerosol distribution. To address potential variability in star irradiance, a dynamic method is applied to determine the log(USI0) value at each moment, which helps minimize the influence of intrinsic star irradiance variability.

As for the spectral response, the manuscript partially addresses this issue by evaluating a second camera model (OMEA-3C-TF) equipped with narrower-band filters. This allows us to assess the influence of spectral bandwidth on the accuracy of the retrieved AOD values.

Finally, the performance and precision of the proposed method have been evaluated using co-located measurements from moon photometers as reference, thereby quantifying the reliability of the system under real observational conditions.

***Methodological and Calibration Challenges***

RC: Strengths:
  - **Data Extraction:** Extracting starlight signals from wide-field images is technically innovative and promising for remote sensing.
Analysis:
  - **Calibration:** All-sky cameras are not designed for absolute radiometry, requiring robust calibration strategies.
  - **Data Processing:** Background subtraction, optical corrections, and noise propagation need clearer methodological details.

**- Spectral Considerations:** The study should address how it handles spectral mismatches affecting AOD retrieval.

AC: One advantage of this method is that absolute calibration is not required, thanks to the use of the Langley technique. We believe that a large part of the methodology is explained in considerable detail in the article, such as in the case of background subtraction, where we specify which Python libraries were used and with which input parameters.

Although the error propagation is not described in full detail, the manuscript does state which sources of uncertainty are propagated and that this propagation is done quadratically. In our opinion, going further into these aspects would make the article harder to read by introducing overly technical information.

**Data Validation & Comparative Analysis**

**RC: Strengths:**
- **Preliminary Validation:** Initial consistency with existing measurements suggests potential feasibility.

**Analysis:**
- **Systematic Biases:** Possible AOD overestimation needs deeper investigation.
- **Comparative Studies:** A broader validation campaign against reference instruments across various aerosol regimes is necessary.
- **Temporal & Spatial Variability:** The method's ability to resolve short-term and spatial variations requires further evaluation.

AC: We fully agree that further investigation is needed to better understand the causes of the observed overestimation in AOD. Although this study presents an extensive comparison against lunar photometer data (covering nine different sites around the world with varying latitudes and aerosol conditions), we also acknowledge the need for a broader validation effort in the future. This would not only allow us to address a wider range of aerosol regimes but also to compare results with other instruments, such as star photometers. This would be especially valuable for assessing the performance of the proposed methodology during moonless nights. This need has been reflected in the conclusions section of the manuscript.

**Radiative Transfer Modeling & Algorithmic Considerations**

**RC: Strengths:**
- **Model Integration:** Radiative transfer models provide a strong theoretical basis.

**Analysis:**
- **Model Assumptions:** Question: Should the impact of multiple scattering and horizon effects be analyzed?.
- **Algorithm Robustness:** Star identification and retrieval parameter sensitivity require systematic validation.

AC: The impact of multiple scattering should be considered, as with any other photometric instrument. However, in our case, since only a few pixels corresponding to the star are selected, the effective field of view is relatively small, and thus this effect should not be too significant. Additionally, multiple scattering is expected to play a more relevant role under high aerosol loads and in the presence of coarse and super coarse particles, which are not the most frequent conditions.

Regarding the effect of the horizon, it is true that certain lights in the skyline of some sites may introduce additional brightness in the sky. While the background correction algorithm is generally able to remove this effect, in some stations these lights can saturate portions of the

image. To minimize possible issues near the horizon, stars with zenith angles greater than 80° were excluded. Nevertheless, the impact of these factors (multiple scattering and horizon) should be analyzed in future studies, but we think this lies beyond the scope of the present work.

As for the star identification algorithm, it has been found to work with high accuracy, provided that the geometric calibration of the camera is correctly performed. It has also been observed that, in some rare cases, the algorithm does not detect the target star and instead selects a nearby brighter one. To address this, outlier detection algorithms were implemented as described in the manuscript. This helps filtering out such cases, since the signal from a misidentified star will deviate from expectations and will not pass the applied filters.

**_Operational Considerations & Generalization_**

**RC: Strengths:**
- **Scalability:** The approach could enhance nighttime aerosol monitoring at low cost!

**Analysis:**
- **Environmental Dependence:** The influence of clouds, light pollution, and geographic variability should be better addressed.

  **Hardware Variability:** Standardized calibration across different camera models and long-term stability assessments are necessary.

AC: Cloud filtering is definitely one of the key challenges to be addressed in future work, as mentioned in the conclusions of the manuscript. In this study, we worked with two very similar camera models, and the proposed methodology appears to yield promising results. We consider this work to be a first step toward extending the approach to a wider range of camera types in the future. In fact, we are currently working on adapting this methodology to other camera models. We also believe that this study highlights the importance of being able to record raw images from all-sky cameras, as well as the benefits of using narrower spectral filters.

**_Future Directions & Recommendations_**

**RC:**
- Improve calibration for sensor aging and optical distortions.
- Quantify uncertainties using advanced statistical methods.
- Enhance star detection and background subtraction algorithms.
- Participate or organize more validation campaigns under diverse conditions.

AC: We appreciate these recommendations, which we fully agree should guide future work. In fact, most of them are aligned with the future directions already outlined in the conclusions section of the manuscript.

**_Conclusion_**

**RC**: This study presents a very promising, cost-effective method for nighttime aerosol monitoring. Refining the calibration, the validation, and maybe the error analysis will be crucial to ensuring its reliability and broader adoption in atmospheric research.

AC: We would like to thank the reviewer for this comment. While there are still several aspects of the methodology that can be further refined, particularly in terms of quality data filtering, cloud screening, and more effective calibration, we believe this work represents a first step towards the systematic use of cost-effective instruments such as all-sky cameras for night-time AOD retrieval.

**Response to the Anonymous Referee #3 comments for the manuscript "Star photometry with all-sky cameras to retrieve aerosol optical depth at night-time" By Roberto Román et al. in AMT**

First of all, we would like to thank the time and effort of the referee for the detailed review of the manuscript. Reviewer comments (RC) are in black font and author comments (AC) are in red.

**Author's answer to Anonymous Referee #3**

**General comments**
RC: The manuscript is about a relevant topic in aerosol remote sensing, specifically exploring the method of star photometry with all-sky cameras. The method yields interesting results, the text is well written, and so from my point of view, it is suitable for publication in AMT. Below, I have added some comments, and I leave it to the authors to decide if/ how to consider them for the manuscript.
AC: We appreciate very much this comment from the reviewer.

**Specific comments:**
RC: L272, Eqn. 2. It's probably pretty standard, so it could be omitted in case the manuscript needed to be streamlined.
AC: We agree that this equation is well known. It has been removed in the revised version of the manuscript.

RC: L382pp. While you can of course use a RT model, for the direct irradiance it would be a bit of an overkill, when you can use a simple equation (as in the Beer-Lambert law)?.
AC: We agree that for estimating direct irradiance under these conditions, the Beer-Lambert law may suffice. However, although the use of a complex radiative transfer model may not be necessary for this task, we opted to use the libRadtran package because it was more practical for us, given our experience with its use. Furthermore, this approach allowed us to incorporate the concentrations of gases in a standard atmosphere and the extinction coefficients for different gases at each wavelength, which are already included in the libRadtran package. This facilitated the calculation in a simpler way for us, and ultimately, it is as valid a method as directly using the Beer-Lambert law.

RC: By the way, also try to avoid the common confusion with libRadtran: it is not a "model" but a software package that includes multiple models (or solvers) like DISORT and MYSTIC (Monte Carlo code).
AC: Totally agree. We have rewritten the sentence as next:

"*To calculate $\lambda_{eff}$, and to ensure realistic values of direct irradiances, the spectral direct transmittance of the Earth's atmosphere is simulated under different conditions using the DISORT radiative transfer model (Stamnes et al., 1988; Buras et al., 2011) included in the libRadtran 2.0.4 radiative transfer package (Mayer and Kylling, 2005; Emde et al., 2016)*".

RC: L417pp. Again, is it probably not necessary to use a RT model here? Basically it's about the cross sections that you use, no? Also, while Fig. 7 is of course relevant in general, it is maybe more appropriate in a textbook about remote sensing. To me it seems a bit distracting in a paper focusing on a specific implementation of star photometry with a camera.

AC: As in the previous case, it is not strictly necessary to use an RT model, but it was more practical and easier for us, and we were able to take advantage of the cross-section coefficients that are included in the package.

Regarding Figure 7, it might be the one that least fits with the rest of the figures in the article. However, we believe that the method proposed for retrieving the optical depth of gases (including Rayleigh) is somewhat complex, particularly when dealing with spectral bands broader than usual. Therefore, we consider that this figure can help the reader better understand the method. That is the reason why we have chosen to keep it in the manuscript for now.

RC: Fig. 9. I could think that there are more interesting or representative cases to be shown?.
AC: The reviewer is correct in noting that there are more interesting cases. But while it is true that there may be other, more interesting cases, this particular case was chosen somewhat arbitrarily. We believe that there are likely more representative days, but these are already addressed in Section 4.2, which focuses on case studies. In the case of Figure 9, the main interest lies in showing how the AOD remains relatively stable during the night and discussing how and why some data points do not pass the established filters.

RC: Fig. 10. I found myself a couple of times trying to relate camera number and location (for simplicity sometimes only number or location is used in the discussion). I know, the figure is quite busy already with text and numbers, but maybe there is a way to include the locations or generally have a more efficient labeling?
AC: We are aware that the labeling can sometimes be somewhat confusing due to the large number of used cameras and in several locations, with some cameras being in multiple sites and some sites having multiple cameras. While we could separate the data by location, this would require mixing data from two cameras in Lindenberg and three in Valladolid (with two different camera models in the latter case). Alternatively, we could separate by cameras and subdivide by location, for example, by separating C005 in Valladolid, Izaña, Fuencaliente, and Andoya, as shown in Tables 5 and 6. However, this would result in too many panels in Figure 10, which already contains a significant number. Moreover, creating a new figure with this subdivision could be redundant, as most of the information is already provided in Tables 5 and 6.

Despite the potential confusion, and in order to avoid mixing data from different cameras, we believe that the current way of presenting the results is the most appropriate.

RC: L563pp. So how exactly would an error in the Langley show in the correlation scatter plot? Considering the airmass dependence of the AOD error it would cause some scatter, but only in one direction (above or below the 1:1 line)? If yes, would it be radiometric instability or sth else that causes the low correlation?
AC: We believe the reviewer has made an insightful observation in this comment. If the constant calibration log(USI0) –ln(USI0) in the new manuscript– obtained through the Langley method, is higher than its true value, then the AOD will always be overestimated, and vice versa. However, in our case, we do not rely on a single star with a unique value of log(USI0), but instead, the AOD is obtained at each moment by averaging the AOD values from different stars. These individual AOD values may be either overestimated or underestimated if the value of log(USI0) is not perfectly calculated, which is more complicated in high-latitude stations when using the Langley method. This is because the optical mass variation is slower in those locations, making it take longer to perform a Langley, and consequently, there is a higher likelihood of atmospheric changes during that longer period.

Therefore, the final AOD is an average of AODs that may be overestimated, underestimated, or well estimated. However, the individual AODs available change throughout the night and also during the year. As a result, we can have situations where the AOD could be sometimes overestimated and other times underestimated due to inaccuracies in the calibration constant (logUSI0) of the individual stars.